# On Convergence Rates of Deep Nonparametric Regression under Covariate Shift

## Abstract

Traditional machine learning and statistical modeling methodologies are rooted in a fundamental assumption: that both training and test data originate from the same underlying distribution. However, the practical reality often presents a challenge, as training and test distributions frequently manifest discrepancies or biases. In this work, we study covariate shift, a type of distribution mismatches, in the context of deep nonparametric regression. We thus formulate a two-stage pre-training reweighted framework relying on deep ReLU neural networks. We rigorously establish convergence rates for the unweighted, reweighted, and pre-training reweighted estimators, illuminating the pivotal role played by the density-ratio reweighting strategy. Additionally, our analysis illustrates the efficacy of pre-training and provides valuable insights for practitioners, offering guidance for the judicious selection of the number of pre-training samples.

## 1 Introduction

Covariate shift (Quinoñero-Candela et al., 2009; Sugiyama & Kawanabe, 2012), a pervasive phenomenon within the domains of machine learning and statistical modeling, bears notable relevance across diverse domains, including computer vision, natural language processing, and medical image analysis, among others. It distinguishes itself from conventional machine learning and statistical modeling paradigms, where the traditional assumption posits that both training and testing data originate from the same underlying distribution. However, covariate shift manifests during the modeling process when the distribution of training data significantly deviates from that of the testing data. In simpler terms, covariate shift represents a scenario wherein the statistical properties of the data undergo substantial changes between the training and testing phases of a machine learning or statistical model. This phenomenon often leads to a deterioration in the model's generalization capability, as the model has primarily learned patterns from one distribution but is then tasked with making predictions on a distinctly different distribution. Consequently, many researchers have proposed an array of methods to address this intricate issue, seeking to mitigate the adverse effects of covariate shift on model performance. In the work of Kuang et al. (2021), a balanced-subsampled stable prediction algorithm is proposed, which is based on the fractional factorial design strategy. This algorithm is designed to address covariate balancing and ensure stable predictions. Duchi & Namkoong (2021) presented a distributionally robust stochastic optimization framework aimed at mitigating distribution shifts. It leverages the concept of $f$-divergence (Csiszár, 1967) to quantify the magnitude of distribution shift from the training distribution. Subsequently, the framework is implemented by incorporating the empirical distribution plug-in methodology. Krueger et al. (2021) introduced the risk extrapolation method, which performs as a form of robust optimization applied across a perturbation set that encompasses extrapolated domains. Additionally, they have devised a penalty function to tackle the variance in training risks, providing a simplified alternative. Dubois et al. (2021) explored covariate shift by focusing on the acquisition of optimal representations, ensuring that source risk minimizers generalize effectively across distributional shifts. It is paramount to underscore that density-ratio reweighting, referred to as importance weighting (Shimodaira, 2000; Huang et al., 2006; Sugiyama & Storkey, 2006; Sugiyama et al., 2007a;b; Bickel et al., 2007; Sugiyama et al., 2008; Bickel et al., 2009; Kanamori et al., 2009; Fang et al., 2020), emerges as a primary approach for addressing the intricacies of covariate shift. Some researchers have conducted extensive error analysis related to ratio-reweighting, as documented in the works of Cortes et al. (2008; 2010); Xu et al. (2022). Recently, Ma et al. (2023) has explored the covariate shift problem within the frame-

work of nonparametric regression over a reproducing kernel Hilbert space (RKHS), and endeavored to provide some theoretical insights. In these studies, the authors assume prior knowledge of the test distribution and directly employ the exact density ratio for the theoretical analysis. However, in practical scenarios, the precise density ratio is often unattainable. In this paper, we propose a pre-training strategy to initially estimate the density ratio using unlabeled data from both the source and target distributions, followed by the derivation of the pre-training reweighted estimator. Furthermore, covariate shift is also closely intertwined with out-of-distribution generalization, transfer learning, domain adaptation, and stable learning. However, due to space constraints, we reserve this discussion for Appendix A.

In comparison to RKHS, deep neural networks (DNNs) also stand out as a formidable technique employed within machine learning and statistics. The capabilities of DNNs extend far beyond their superficial appearance, encompassing a wide spectrum of applications and possibilities (Goodfellow et al., 2016). Recently, DNNs have catalyzed a surge of interest in the field of deep nonparametric regression, wherein these neural networks are harnessed to approximate underlying regression functions within the context of nonparametric regression (Stone, 1982; Gyorfi et al., 2002; Tsybakov, 2009). Numerous studies have contributed to our understanding of deep nonparametric regression. Remarkable works by Bauer & Kohler (2019); Schmidt-Hieber (2020); Nakada & Imaizumi (2020); Kohler & Langer (2021); Farrell et al. (2021); Chen et al. (2022); Kohler et al. (2022); Nakada & Imaizumi (2022); Jiao et al. (2023), and others have illuminated various facets of this burgeoning field, elucidating intricate theoretical properties and unveiling innovative methodologies. However, it is noteworthy that existing literature has thus far overlooked the covariate shift phenomenon inherent in deep nonparametric regression. Building upon this research trajectory, our investigation embarks on a meticulous exploration of the covariate shift phenomenon in nonparametric regression utilizing DNNs. To the best of our knowledge, this work conducts the first effort to uncover and provide robust theoretical guarantees pertaining to the covariate shift phenomenon within the context of deep nonparametric regression.

## 1.1 CONTRIBUTIONS

Covariate shift poses a significant challenge in practical applications. One effective strategy for mitigating the impact of covariate shift involves the utilization of a ratio-reweighted approach within the context of nonparametric regression. However, it is important to note that in practical scenarios, the precise density ratio is often unattainable, leaving us with access to only unlabeled data from both the source and target distributions. To address this challenge, we propose a two-stage neural network-based methodology tailored to this specific scenario. In the initial pre-training stage, we leverage unlabeled data from both the source and target distributions to estimate the density ratio through deep logistic regression. Subsequently, we seamlessly integrate a ratio-reweighted strategy into the framework of deep nonparametric regression, relying exclusively on labeled data originating from the source distribution. To underpin the effectiveness and robustness of our approach, we provide a comprehensive error analysis encompassing unweighted, reweighted, and pre-training-reweighted estimators. In summary, the main contributions of this study can be outlined as follows.

(i) We present oracle inequalities for unweighted, reweighted, and pre-training-reweighted estimators. These inequalities are derived through a decomposition of the excess risk into two terms: approximation error and generalization error (statistical error). Additionally, the established generalization error bounds exhibit fast rates $\mathcal{O}(1/n)$, demonstrating a substantial enhancement over the slow rate $\mathcal{O}(1/\sqrt{n})$ originally posited by Cortes et al. (2008; 2010).

(ii) By strategically balancing these two terms, we establish convergence rates for the aforementioned estimators. These obtained convergence rates are with respect to the boundedness of the density ratio and the sample size, which explicitly account for the impact of the shift between source and target distributions. To the best of our knowledge, we are the first to elucidate the convergence rates of deep nonparametric regression in the presence of covariate shift. Moreover, comparing these rates between unweighted and reweighted estimators provides strong evidence of the benefits of ratio reweighting.

(iii) The convergence rates of the pre-training-reweighted estimator reflect the combined influence of the number of unlabeled samples utilized in the pre-training phase and the number of labeled samples employed in subsequent deep nonparametric regression, where the rates of the density ratio estimator attain minimax optimal rates. These theoretical results provide valuable insights

for determining not only the sample size for nonparametric regression, but also the sample size required for the pre-training procedure.

## 1.2 MAIN RESULTS

In this section, we expound upon the primary findings of this paper, specifically, the convergence rates pertaining to the unweighted estimator, reweighted estimator, and pre-training reweighted estimator within the context of the target distribution.

| Estimator | Definition | Extra Information | Convergence Rates | |
|---|---|---|---|---|
| Unweighted estimator | (2.3) | None | $\mathcal{O}(\Lambda n^{-\frac{2\beta}{d+2\beta}})$ | Theorem 3.4 |
| Reweighted estimator | (2.5) | Density ratio | $\mathcal{O}(\Lambda^{\frac{2\beta}{d+2\beta}} n^{-\frac{2\beta}{d+2\beta}})$ | Theorem 3.8 |
| Pre-training reweighted estimator | (2.9)(2.10) | Unlabeled data | $\mathcal{O}(\Lambda^{\frac{2\beta}{d+2\beta}} n^{-\frac{2\beta}{d+2\beta}}) + \mathcal{O}(m^{-\frac{\alpha}{d+2\alpha}})$ | Theorem 3.13 |

## 1.3 PRELIMINARIES AND NOTATIONS

In this section, we introduce the notations utilized throughout this paper and subsequently present the definition of ReLU DNNs.

Let $P$ be a joint distribution over $\mathcal{X} \times \mathcal{Y}$. Denote by $P_X$ the marginal distribution of $X$ and $P_{Y|X}$ the conditional distribution of $Y$ given $X$. By a similar argument, we can define the marginal distribution $Q_X$ and conditional distribution $Q_{Y|X}$ for some joint distribution $Q$. Then by the definition it holds that $P(X, Y) = P_{Y|X}(Y|X)P_X(X)$ and $Q(X, Y) = Q_{Y|X}(Y|X)Q_X(X)$. Denote by $\mathscr{L}(\mathcal{X})$ the set of measurable functions on $\mathcal{X}$.

**Definition 1.1** (ReLU DNNs). A neural network $\psi : \mathbb{R}^{N_0} \to \mathbb{R}^{N_{L+1}}$ is a function defined by

$$\psi(x) = T_L(\phi(T_{L-1}(\cdots \phi(T_0(x))\cdots))),$$

where the ReLU activation function $\phi(x) := \max\{0, x\}$ is applied component-wisely and $T_\ell(x) := A_\ell x + b_\ell$ is an affine transformation with $A_\ell \in \mathbb{R}^{N_{\ell+1} \times N_\ell}$ and $b_\ell \in \mathbb{R}^{N_\ell}$ for $\ell = 0, \ldots, L$. In this paper, we consider the case $N_0 = d$ and $N_{L+1} = 1$. The numbers $W := \max\{N_1, \ldots, N_L\}$ and $L$ are called the width and the depth of neural networks, respectively. Additionally, $S := \sum_{\ell=0}^{L} N_\ell N_{\ell+1} \leq LW^2$ is called the number of parameters of the neural network. We denote by $\mathcal{N}(W, L)$ the set of functions implemented by ReLU neural networks with width at most $W$ and depth at most $L$.

## 1.4 ORGANIZATION

The remainder of this manuscript is structured as follows. Section 2 provides a comprehensive exposition of the formulation of the two-stage pre-training-reweighted algorithm. It involves in deriving three distinct estimators: the unweighted estimator, the reweighted estimator, and the pre-training reweighted estimator. Section 3 furnishes the convergence analysis of these estimators, elucidating their respective rates. Section 4 summarizes the conclusions of this work. Appendix A presents a review of related work. Appendices B to E provides comprehensive technical proofs for all the lemmas and theorems presented in this paper.

## 2 PROBLEM FORMULATION

Let $\mathcal{X} \subseteq [0, 1]^d$ ($d \geq 1$) be the feature space and $\mathcal{Y} \subseteq \mathbb{R}$ be the response space. Consider the following nonparametric regression model

$$Y = f_0(X) + \xi. \tag{2.1}$$

Here, the response variable $Y \in \mathcal{Y}$ is associated with the covariate $X \in \mathcal{X}$. The underlying regression function is defined as $f_0 : \mathcal{X} \to \mathbb{R}$. Furthermore, $\xi$ represents a random noise term that is independent of $X$ and satisfies the condition $\mathbb{E}[\xi] = 0$. It is obvious that $f_0(x) = \mathbb{E}[Y|X = x]$ for each $x \in \mathcal{X}$. In particular, our analysis focus on cases where the noise term $\xi$ exhibits sub-Gaussian characteristics shown in the following assumption.

**Assumption 1** (Sub-Gaussian noise). The noise $\xi$ in (2.1) is a sub-Gaussian random variable with mean 0 and finite variance proxy $\sigma^2$, that is, its moment generating function satisfies

$$\mathbb{E}[\exp(a\xi)] \leq \exp\left(\frac{\sigma^2 a^2}{2}\right), \quad \forall\, a \in \mathbb{R}.$$

In the context of nonparametric regression, our focus centers on the observation of $n$ independent and identically distributed (i.i.d.) random pairs denoted as $\mathcal{D} := \{(X_i^P, Y_i^P)\}_{i=1}^n$. These pairs are drawn from a training distribution $P$ defined over $\mathcal{X} \times \mathcal{Y}$. Specifically, $\{X_i^P\}_{i=1}^n$ are sampled from $P_X$, and the conditional probability $P_{Y|X}$ is determined in accordance with (2.1). We introduce another distribution $Q$ as the test distribution over $\mathcal{X} \times \mathcal{Y}$. More precisely, the covariates of test samples are derived from $Q_X$, and the corresponding responses are also generated following the model (2.1), implying $Q_{Y|X} = P_{Y|X}$.

Within this context, $P_X$ is referred to as the source distribution for covariates, while $Q_X$ serves as the target distribution for covariates. In practical scenarios, it is common for the target distribution $Q_X$, on which a model is deployed, to exhibit divergence from the source distribution $P_X$. This phenomenon is commonly referred to as covariate shifts, and it can be attributed to various factors, including temporal or spatial data evolution or inherent biases introduced during the data collection process.

Our primary objective revolves around the development of an estimator denoted as $\hat{f}$, which is constructed based on the observed data. This estimator is carefully designed with the primary aim of minimizing the $L^2(Q_X)$-risk, as defined below:

$$\|\hat{f} - f_0\|_{L^2(Q_X)}^2 = \mathbb{E}_{X \sim Q_X}\left[(\hat{f}(X) - f_0(X))^2\right] = \int_{\mathcal{X}} (\hat{f}(x) - f_0(x))^2 q_X(x) dx, \quad (2.2)$$

where $q_X$ represents the probability density function associated with $Q_X$. In essence, our goal is to minimize this risk, which quantifies the expected squared difference between the estimator $\hat{f}$ and the underlying regression function $f_0$ over the distribution $Q_X$.

## 2.1 Unweighted Estimators

When confronted with a situation where information about the target distribution is unavailable, a natural approach for constructing an estimator is to directly minimize the unweighted empirical risk over a hypothesis class $\mathcal{F}$, representing a set of measurable functions. This estimator, denoted as $\hat{f}_{\mathcal{D}}$, is determined as follows:

$$\hat{f}_{\mathcal{D}} \in \arg\min_{f \in \mathcal{F}} \widehat{L}_{\mathcal{D}}(f) := \frac{1}{n} \sum_{i=1}^n (f(X_i^P) - Y_i^P)^2. \quad (2.3)$$

It is important to note that $\widehat{L}_{\mathcal{D}}(f)$ serves as a sample average approximation to the unweighted population risk $L(f)$, expressed as:

$$L(f) := \mathbb{E}_{(X^P, Y^P) \sim P}\left[(f(X^P) - Y^P)^2\right].$$

According to (2.1), it is straightforward to verify that $L(\hat{f}_{\mathcal{D}}) = \|\hat{f}_{\mathcal{D}} - f_0\|_{L^2(P_X)}^2 + \sigma^2$, which means that the minimizer of the unweighted population risk concurrently minimizes the $L^2(P_X)$-risk, as opposed to $L^2(Q_X)$-risk defined in (2.2).

## 2.2 Reweighted Estimators

When we have knowledge of the target distribution, a direct approach is available for minimizing the population risk with the exact weight. To facilitate this, we introduce the concept of the density ratio, as defined for the target distribution $Q_X$ and the source distribution $P_X$. We denote by $p_X$ the probability density function of $P_X$, then this density ratio, denoted as $\varrho(x) := q_X(x)/p_X(x)$, is also referred to as the importance weight (Cortes et al., 2010). It is worth noting that the density ratio $\varrho(\cdot)$ measures the discrepancy between $Q_X$ and $P_X$. In this work, we specifically consider the scenario where the density ratio is uniformly upper-bounded, a condition we formalize as follows:

**Assumption 2** (Uniformly upper-bounded density ratio). The density ratio $\varrho$ has a finite upper bound, that is, $\Lambda := \sup_{x \in \mathcal{X}} \varrho(x) < \infty$.

With the exact density ratio $\varrho$ at our disposal, our objective turns to minimizing the population reweighted risk, defined as:

$$L_\varrho(f) := \mathbb{E}_{(X^P, Y^P) \sim P} \big[ \varrho(X^P)(f(X^P) - Y^P)^2 \big]. \tag{2.4}$$

Minimizing this population reweighted risk is equivalent to minimizing the $L^2(Q_X)$-risk, that is, $L_\varrho(f) = L_Q(f)$. A commonly employed approach to achieve this is through empirical reweighted risk minimization within a hypothesis class $\mathcal{F}$, resulting in the following estimator:

$$\hat{f}_{\varrho, \mathcal{D}} \in \arg\min_{f \in \mathcal{F}} \widehat{L}_{\varrho, \mathcal{D}}(f) := \frac{1}{n} \sum_{i=1}^{n} \varrho(X_i^P)(f(X_i^P) - Y_i^P)^2. \tag{2.5}$$

This approach leverages the density ratio $\varrho$ to reweight the contributions of individual samples in the empirical risk minimization, effectively adapting the learning process to account for the covariate shift.

## 2.3  PRE-TRAINING REWEIGHTED ESTIMATORS

However, the accessibility of precise density ratio functions frequently confronts inherent limitations in practical applications. Nonetheless, it is pertinent to note that a pragmatic solution presents itself in the form of deriving estimations for these density ratio functions. This can be achieved through the utilization of unlabeled data from both the source and target distributions. The methodology underpinning this estimation process aligns with the well-defined principles expounded upon in the following lemma.

**Lemma 2.1.** *Let $p_X$ and $q_X$ be two probability density functions on $\mathcal{X}$. Then the density ratio $\varrho$ is given by $\varrho(x) = q_X(x)/p_X(x) = \exp(-u^*(x))$, where the function $u^*$ satisfies*

$$u^* = \arg\min_{u \in \mathscr{L}(\mathcal{X})} \Big\{ \mathbb{E}_{X \sim P_X} \big[ \log(1 + \exp(-u(X))) \big] + \mathbb{E}_{X \sim Q_X} \big[ \log(1 + \exp(u(X))) \big] \Big\}. \tag{2.6}$$

*Remark* 2.2 (Pseudo-labels). Let $X^P$ and $X^Q$ be random variables distributed from $P_X$ and $Q_X$, respectively. We assign a pseudo-label $Z^P = +1$ for $X^P$ and $Z^Q = -1$ for $X^Q$, and construct a random variable pair $(X^\mu, Z^\mu)$ by

$$\begin{cases} X^\mu = \sigma X^P + (1 - \sigma) X^Q, \\ Z^\mu = \sigma Z^P + (1 - \sigma) Z^Q, \end{cases} \tag{2.7}$$

where $\sigma$ is a random variable satisfying $\Pr(\sigma = 1) = \Pr(\sigma = 0) = 1/2$ and is independent of $X^P, Z^P, X^Q, Z^Q$. We denote by $\mu$ the joint distribution of $(X^\mu, Z^\mu)$ in (2.7), then the population logistic risk can be given by $L_{\text{logit}}(u) = \mathbb{E}_{(X^\mu, Z^\mu) \sim \mu}[\log(1 + \exp(-Z^\mu u(X^\mu)))]$, which is the objective function in (2.6).

As shown in Lemma 2.1, we define the population pre-training risk as

$$L^{\text{pre}}(u) = \mathbb{E}_{X^P \sim P_X} \big[ \log(1 + \exp(-u(X^P))) \big] + \mathbb{E}_{X^Q \sim Q_X} \big[ \log(1 + \exp(u(X^Q))) \big].$$

Let $\mathcal{S}_P := \{X_i^P\}_{i=1}^m$ and $\mathcal{S}_Q := \{X_i^Q\}_{i=1}^m$ represent the collections of unlabeled samples originating from $P_X$ and $Q_X$, respectively. We proceed by assigning pseudo-labels $Z_i^P = +1$ for $X_i^P$ and $Z_i^Q = -1$ for $X_i^Q$. Accordingly, we can construct a pseudo-labeled sample set $\mathcal{S} := \{(X_i^\mu, Z_i^\mu)\}_{i=1}^m$ following the expressions:

$$\begin{cases} X_i^\mu = \sigma_i X_i^P + (1 - \sigma_i) X_i^Q, \\ Z_i^\mu = \sigma_i Z_i^P + (1 - \sigma_i) Z_i^Q, \end{cases} \tag{2.8}$$

where $\{\sigma_i\}_{i=1}^m$ are random variables satisfying $\Pr(\sigma_i = 1) = \Pr(\sigma_i = 0) = 1/2$. Consequently, the empirical pre-training risk $\widehat{L}_{\mathcal{S}}^{\text{pre}}(\cdot)$ is formulated as:

$$\widehat{L}_{\mathcal{S}}^{\text{pre}}(u) := \frac{1}{m} \sum_{i=1}^{m} \log(1 + \exp(-Z_i^\mu u(X_i^\mu))).$$

It is straightforward to verify that $\mathbb{E}_{\mathcal{S}}\widehat{L}_{\mathcal{S}}^{\mathrm{pre}}(u) = L^{\mathrm{pre}}(u)$ for each fixed $u \in \mathscr{L}(\mathcal{X})$. Subsequently, the minimization of the empirical pre-training risk $\widehat{L}_{\mathcal{S}}^{\mathrm{pre}}(\cdot)$ within a given hypothesis class $\mathcal{U}$ yields the following density ratio estimator:

$$\hat{\varrho}_{\mathcal{S}} = \exp(-\hat{u}_{\mathcal{S}}), \quad \text{where } \hat{u}_{\mathcal{S}} \in \arg\min_{u \in \mathcal{U}} \widehat{L}_{\mathcal{S}}^{\mathrm{pre}}(u). \tag{2.9}$$

By substituting the weight function $\varrho$ in (2.4) and (2.5) with $\hat{\varrho}_{\mathcal{S}}$, we derive the population pre-training-reweighted risk, defined as follows:

$$L_{\hat{\varrho}_{\mathcal{S}}}(f) := \mathbb{E}_{(X^P, Y^P) \sim P}\big[\hat{\varrho}_{\mathcal{S}}(X^P)(f(X^P) - Y^P)^2\big].$$

Then, the pre-training-reweighted estimator is formulated as follows:

$$\hat{f}_{\hat{\varrho}_{\mathcal{S}}, \mathcal{D}} \in \arg\min_{f \in \mathcal{F}} \widehat{L}_{\hat{\varrho}_{\mathcal{S}}, \mathcal{D}}(f) = \frac{1}{n}\sum_{i=1}^{n} \hat{\varrho}_{\mathcal{S}}(X_i^P)(f(X_i^P) - Y_i^P)^2. \tag{2.10}$$

We notice that in the first stage, $\mathcal{S} = \{(X_i^\mu, Z_i^\mu)\}_{i=1}^m$, is independent of the second stage data, $\mathcal{D} = \{(X_i^P, Y_i^P)\}_{i=1}^n$. This independence between the two stages is noteworthy.

In summary, we present the pre-training-reweighted algorithm designed for regression tasks under the influence of covariate shift, as outlined in Algorithm 1.

---

**Algorithm 1** Two-stage life-cycle of pre-training-reweighted regression under covariate shift.

**Input:** Unlabeled data for pre-training
- $\mathcal{S}_P = \{X_i^P\}_{i=1}^m$: Unlabeled data sampled from the source distribution $P_X$.
- $\mathcal{S}_Q = \{X_i^Q\}_{i=1}^m$: Unlabeled data sampled from the target distribution $Q_X$.

1: **Pre-training: Estimate density ratio.**
2: Construct pseudo-labeled sample set $\mathcal{S} = \{(X_i^\mu, Z_i^\mu)\}_{i=1}^m$ by (2.8).
3: Minimize the empirical logistic risk based on $\mathcal{S}$:

$$\hat{u}_{\mathcal{S}} \in \arg\min_{u \in \mathcal{U}} \frac{1}{m}\sum_{i=1}^m \log(1 + \exp(-Z_i^\mu u(X_i^\mu))).$$

**Output:** The density ratio estimator $\hat{\varrho}_{\mathcal{S}} = \exp(-\hat{u}_{\mathcal{S}})$.
**Input:** The density ratio estimator $\hat{\varrho}_{\mathcal{S}}$ and labeled data $\mathcal{D} = \{(X_i^P, Y_i^P)\}_{i=1}^n$ sampled from $P$.

4: **Reweighted regression.**
5: Minimize the empirical pre-training-reweighted risk based on $\mathcal{D}$:

$$\hat{f}_{\hat{\varrho}_{\mathcal{S}}, \mathcal{D}} \in \arg\min_{f \in \mathcal{F}} \frac{1}{n}\sum_{i=1}^n \hat{\varrho}_{\mathcal{S}}(X_i^P)(f(X_i^P) - Y_i^P)^2.$$

**Output:** The pre-training-reweighted estimator $\hat{f}_{\hat{\varrho}_{\mathcal{S}}, \mathcal{D}}$.

---

This algorithm is structured into two stages, each serving a crucial role in mitigating the challenges posed by covariate shift. The first stage, denoted as the unsupervised pre-training stage, involves the generation of pseudo-labels $\{Z_i^\mu\}_{i=1}^m$ for unlabeled data $\{X_i^\mu\}_{i=1}^m$, as per the methodology detailed in (2.8). Additionally, in this stage, we estimate the density ratio $\varrho$ by employing logistic regression on the unlabeled data augmented with these pseudo-labels. In the second stage, the supervised phase, we employ the pre-trained density ratio $\hat{\varrho}_{\mathcal{S}}$ in conjunction with the labeled dataset $\mathcal{D} = \{(X_i^P, Y_i^P)\}_{i=1}^n$ to estimate the underlying regression function $f_0$. This multi-stage approach encapsulates the essence of the pre-training-reweighted algorithm, which strategically combines unsupervised and supervised learning paradigms to address the challenges posed by covariate shift in regression tasks.

## 3 CONVERGENCE ANALYSIS

Up to now, we have introduced three estimators: unweighted estimator (2.3), reweighted estimator (2.5), and pre-training-reweighted estimator (2.10). Our objective in this section is to demystify the

advantage of ratio-reweighting and pre-training in deep nonparametric regression under covariate shift. To accomplish this, we will rigorously analyze the performance of these three estimators by addressing the following fundamental questions:

> *What are convergence rates of three estimators (2.3), (2.5) and (2.10) in the presence of covariate shift? How do these convergence rates depend on the discrepancy between the source and target distribution?*

Technically, the convergence analysis can be dissected into two fundamental components: the statistical error and the approximation error. Deep approximation theory has played a pivotal role in elucidating the capabilities of neural networks in approximating smooth functions. A comprehensive body of work, encompassing studies by Yarotsky (2018); Yarotsky & Zhevnerchuk (2020); Shen et al. (2019); Shen (2020); Lu et al. (2021); Petersen & Voigtlaender (2018); Jiao et al. (2023), has established the theoretical foundations for understanding the approximation capabilities of deep neural networks. These theoretical contributions have provided insights into the capacity of neural networks to represent complex functions effectively. To bound the statistical error, researchers have harnessed the tools of empirical process theory. The works of Van Der Vaart & Wellner (1996); Van de Geer & van de Geer (2000); Van der Vaart (2000); Bartlett et al. (2005); Giné & Nickl (2021) have been instrumental in this regard. These tools allow for the quantification of statistical errors in terms of the complexity of a hypothesis class, often measured using concepts like the covering number or VC-dimension.

To facilitate our discussion, let us first introduce key findings from the approximation results (Lemma 3.1) presented in Jiao et al. (2023) and delve into the VC-dimension analysis of ReLU neural networks (Lemma 3.2), as explored in Bartlett et al. (2019), and introduce some assumptions (Assumptions 3 and 4). Subsequently, we proceed the convergence rates of these three estimators, as outlined in Sections 3.1 to 3.3. This comprehensive exploration aims to shed light on the intricate dynamics of these estimators in the context of covariate shift, offering valuable insights into their performance and utility.

**Lemma 3.1** (Jiao et al. (2023, Theorem 3.3)). *Let $\mathcal{X} \subseteq [0,1]^d$ and $B > 0$. Let $\beta = s + r$ with $s \in \mathbb{N}$ and $r \in (0,1]$. Assume that $\mu_X$ is absolutely continuous with respect to the Lebesgue measure. For each $U, N \in \mathbb{N}_+$, there exists a ReLU neural network class $\mathcal{N}(W,L)$ with width $W = \mathcal{O}((s+1)^2 d^{s+1} U \log U)$ and depth $L = \mathcal{O}((s+1)^2 N \log N)$ such that*

$$\sup_{f \in \mathcal{H}^\beta(\mathcal{X};B)} \inf_{\psi \in \mathcal{N}(W,L)} \|f - \psi\|_{L^2(\mu_X)}^2 \le cB^2(s+1)^4 d^{2s+\beta \vee 1}(UN)^{-4\beta/d},$$

*where $c$ is an absolute constant.*

**Lemma 3.2** (Bartlett et al. (2019, Theorem 7)). *For each $W, L \in \mathbb{N}_+$, the VC-dimension of a ReLU neural network $\mathcal{N}(W,L)$ with width $W$ and depth $L$ is given by*

$$\mathrm{VCdim}(\mathcal{N}(W,L)) \le cW^2 L^2 \log(WL),$$

*where $c$ is an absolute constant.*

**Assumption 3.** The regression function $f_0$ and functions in hypothesis class $\mathcal{F}$ are bounded, that is, there exists some positive constant $B$ such that $\{f_0\} \cup \mathcal{F} \subseteq \{f : \|f\|_{L^\infty(\mathcal{X})} \le B\}$.

**Assumption 4.** The regression function $f_0$ is Hölder continuous, that is, $f_0 \in \mathcal{H}^\beta(\mathcal{X})$ for some $\beta > 0$.

### 3.1 Unweighted Estimators

**Lemma 3.3** (Oracle inequality of unweighted estimator). *Suppose that Assumptions 1 to 3 hold. Let $\mathcal{D} = \{(X_i^P, Y_i^P)\}_{i=1}^n$ be an i.i.d. sample set drawn from $P$. Suppose that $\mathcal{F}$ is a hypothesis class and $\hat{f}_\mathcal{D} \in \mathcal{F}$ is defined by (2.3). Then the following inequality holds for $n \ge \mathrm{VCdim}(\mathcal{F})$,*

$$\mathbb{E}_{\mathcal{D} \sim P^n}\left[\|\hat{f}_\mathcal{D} - f_0\|_{L^2(Q_X)}^2\right] \lesssim \Lambda \inf_{f \in \mathcal{F}} \|f - f_0\|_{L^2(P_X)}^2 + \Lambda(B^2 + \sigma^2)\mathrm{VCdim}(\mathcal{F})\frac{\log(en)}{n}.$$

**Theorem 3.4** (Convergence rates of unweighted estimator). *Suppose that Assumptions 1 to 4 hold. Assume that $P_X$ is absolutely continuous with respect to the Lebesgue measure. Let*

$\mathcal{D} = \{(X_i^P, Y_i^P)\}_{i=1}^n$ be an i.i.d. sample set drawn from $P$. Set the hypothesis class $\mathcal{F}$ as $\mathcal{F} = \mathcal{N}(W_{\mathcal{F}}, L_{\mathcal{F}})$ with width $W_{\mathcal{F}} = \mathcal{O}(U_{\mathcal{F}} \log U_{\mathcal{F}})$ and depth $L_{\mathcal{F}} = \mathcal{O}(N_{\mathcal{F}} \log N_{\mathcal{F}})$ satisfying $U_{\mathcal{F}} N_{\mathcal{F}} = \mathcal{O}(n^{\frac{d}{2d+4\beta}})$. Suppose $\hat{f}_{\mathcal{D}} \in \mathcal{F}$ is defined by (2.3). Then the following inequality holds

$$\mathbb{E}_{\mathcal{D} \sim P^n}\left[\|\hat{f}_{\mathcal{D}} - f_0\|_{L^2(Q_X)}^2\right] \leq \mathcal{O}\left(\Lambda n^{-\frac{2\beta}{d+2\beta}} (\log n)^2\right).$$

*Remark* 3.5 (Consistency). In Theorem 3.4, the convergence rate is determined as $\mathcal{O}(\Lambda n^{-\frac{2\beta}{d+2\beta}})$, which demonstrates that the prediction error of the unweighted estimator $\hat{f}_{\mathcal{D}}$ is consistent in the sense that $\mathbb{E}\|\hat{f}_{\mathcal{D}} - f_0\|_{L^2(Q_X)}^2 \to 0$ as $n \to \infty$, regardless the $L^\infty$-norm bound $\Lambda$ of density-ratio.

*Remark* 3.6. According to the definition of the density ratio, when $\Lambda = 1$, the scenario simplifies to the standard nonparametric regression. Then, the rate $\mathcal{O}(n^{-\frac{2\beta}{d+2\beta}})$ aligns with the minimax optimal rate within the framework of nonparametric regression, as established in Stone (1982); Gyorfi et al. (2002); Tsybakov (2009). Additionally, our theoretical findings correspond to those in deep nonparametric regression (Schmidt-Hieber, 2020; Kohler & Langer, 2021; Jiao et al., 2023). It is worth noting that our approach differs from the aforementioned literature in terms of the proof methodology employed in statistical error analysis. Specifically, we derive the statistical error primarily based on the offset Rademacher complexity (Liang et al., 2015). For a comprehensive elucidation of our methodology and detailed insights, please refer to Appendices C and D.

## 3.2 REWEIGHTED ESTIMATORS

**Lemma 3.7** (Oracle inequality of exact-reweighted estimator). *Suppose that Assumptions 1 to 3 hold. Let $\mathcal{D} = \{(X_i^P, Y_i^P)\}_{i=1}^n$ be an i.i.d. sample set drawn from $P$. Suppose that $\mathcal{F}$ is a hypothesis class and $\hat{f}_{\varrho, \mathcal{D}} \in \mathcal{F}$ is defined by (2.5). Then the following inequality holds for $n \geq \mathrm{VCdim}(\mathcal{F})$,*

$$\mathbb{E}_{\mathcal{D} \sim P^n}\left[\|\hat{f}_{\varrho, \mathcal{D}} - f_0\|_{L^2(Q_X)}^2\right] \lesssim \inf_{f \in \mathcal{F}} \|f - f_0\|_{L^2(Q_X)}^2 + (\Lambda B^2 + \sigma^2) \mathrm{VCdim}(\mathcal{F}) \frac{\log(e\Lambda n)}{n}.$$

**Theorem 3.8** (Convergence rates of exact-reweighted estimator). *Suppose that Assumptions 1 to 4 hold. Assume that $Q_X$ is absolutely continuous with respect to the Lebesgue measure. Let $\mathcal{D} = \{(X_i^P, Y_i^P)\}_{i=1}^n$ be an i.i.d. sample set drawn from $P$. Set the hypothesis class $\mathcal{F}$ as $\mathcal{F} = \mathcal{N}(W_{\mathcal{F}}, L_{\mathcal{F}})$ with width $W_{\mathcal{F}} = \mathcal{O}(U \log U)$ and depth $L_{\mathcal{F}} = \mathcal{O}(N \log N)$ satisfying $U_{\mathcal{F}} N_{\mathcal{F}} = \mathcal{O}(\Lambda^{-\frac{d}{2d+4\beta}} n^{\frac{d}{2d+4\beta}})$. Suppose $\hat{f}_{\varrho, \mathcal{D}} \in \mathcal{F}$ is defined by (2.5). Then the following inequality holds*

$$\mathbb{E}_{\mathcal{D} \sim P^n}\left[\|\hat{f}_{\varrho, \mathcal{D}} - f_0\|_{L^2(Q_X)}^2\right] \leq \mathcal{O}\left(\Lambda^{\frac{2\beta}{d+2\beta}} (\log \Lambda) n^{-\frac{2\beta}{d+2\beta}} (\log n)^2\right).$$

*Remark* 3.9. The rate $\mathcal{O}(\Lambda^{\frac{2\beta}{d+2\beta}} n^{-\frac{2\beta}{d+2\beta}})$ of the reweighted estimator derived in Theorem 3.8 is much tighter than that $\mathcal{O}(\Lambda n^{-\frac{2\beta}{d+2\beta}})$ of the unweighted estimator in Theorem 3.4, which shows a theoretical advantage of the density-ratio reweighting strategy in nonparametric regression under covariate shift.

## 3.3 PRE-TRAINING REWEIGHTED ESTIMATORS

In this section, we expound upon the cornerstone findings of this paper, focusing on the convergence rate analysis of the pre-training reweighted estimator. This analysis extends beyond the results of the two previous estimators, which makes it distinct from both. Consequently, additional assumptions are introduced, as outlined below (refer to Assumptions 5 to 7). Specifically, one of the most crucial steps involves obtaining density ratio estimator, which are derived from deep nonparametric logistic regression, as detailed in Lemma 3.11. Subsequently, we can ascertain the convergence rate of the pre-training reweighted estimator.

**Assumption 5** (Uniformly lower-bounded density ratio). The density ratio $\varrho$ has a positive lower bound, that is, $\lambda := \inf_{x \in \mathcal{X}} \varrho(x) > 0$.

**Assumption 6.** The log of the density ratio $\varrho$ is Hölder continuous, that is, $\log \varrho \in \mathcal{H}^\alpha(\mathcal{X})$ for some $\alpha > 0$.

**Assumption 7.** For each $u \in \mathcal{U}$, the inequality $\log(1/\Lambda) \leq u(x) \leq \log(1/\lambda)$ holds for each $x \in \mathcal{X}$.

**Lemma 3.10** (Oracle inequality of pre-training-reweighted estimator). *Suppose Assumptions 1 to 3, 5 and 7 hold. Let $\mathcal{S} = \{(X_i^\mu, Z_i^\mu)\}_{i=1}^m$ and $\mathcal{D} = \{(X_i^P, Y_i^P)\}_{i=1}^n$ be two i.i.d. sample sets drawn from $\mu$ and $P$, respectively. Suppose that $\mathcal{U}$ is a hypothesis class and $\hat{u}_\mathcal{S} \in \mathcal{U}$ is defined by (2.9), and suppose that $\mathcal{F}$ is a hypothesis class and $\hat{f}_{\hat{\varrho}_\mathcal{S}, \mathcal{D}} \in \mathcal{F}$ is defined by (2.10). Then the following inequality holds for $m \geq \mathrm{VCdim}(\mathcal{U})$ and $n \geq \mathrm{VCdim}(\mathcal{F})$,*

$$\mathbb{E}_{\mathcal{S} \sim \mu^m} \mathbb{E}_{\mathcal{D} \sim P^n} \left[ \|\hat{f}_{\hat{\varrho}_\mathcal{S}, \mathcal{D}} - f_0\|_{L^2(Q_X)}^2 \right]$$

$$\lesssim B^2 \mathbb{E}_{\mathcal{S} \sim \mu^m} \left[ \|\hat{\varrho}_\mathcal{S} - \varrho\|_{L^2(P_X)} \right] + \inf_{f \in \mathcal{F}} \|f - f_0\|_{L^2(Q_X)}^2 + (\Lambda B^2 + \sigma^2) \mathrm{VCdim}(\mathcal{F}) \frac{\log(e\Lambda n)}{n}.$$

**Lemma 3.11** (Convergence rates of density-ratio estimator). *Suppose that Assumptions 2 and 5 to 7 hold. Assume that $P_X$ and $Q_X$ are absolutely continuous with respect to the Lebesgue measure. Let $\mathcal{S} = \{(X_i^\mu, Z_i^\mu)\}_{i=1}^m$ be an i.i.d. sample set drawn from $\mu$. Set the hypothesis class $\mathcal{U}$ as $\mathcal{U} = \mathcal{N}(W_\mathcal{U}, L_\mathcal{U})$ with width $W_\mathcal{U} = \mathcal{O}(U_\mathcal{U} \log U_\mathcal{U})$ and depth $L_\mathcal{U} = \mathcal{O}(N_\mathcal{U} \log N_\mathcal{U})$ satisfying $U_\mathcal{U} N_\mathcal{U} = \mathcal{O}(n^{\frac{d}{2d+4\alpha}})$. Given $\hat{u}_\mathcal{S} \in \mathcal{U}$ defined in (2.9), then the following inequality holds*

$$\mathbb{E}_{\mathcal{S} \sim \mu^m} \left[ \|\hat{\varrho}_\mathcal{S} - \varrho\|_{L^2(P_X)}^2 \right] \leq \mathcal{O}\left( m^{-\frac{2\alpha}{d+2\alpha}} (\log m)^2 \right).$$

*Remark* 3.12. In Lemma 3.11, we derive the convergence rate of the density-ratio estimator, which is given by $\mathcal{O}(m^{-\frac{2\alpha}{d+2\alpha}})$, with a logarithmic term omitted. The derivation of this error bound is mainly facilitated through the utilization of local complexity techniques (Bartlett et al., 2005), enabling it to achieve the minimax optimal rate.

In view of Theorem 3.8 and Lemmas 3.10 and 3.11, we archive the following rates of convergence of pre-training reweighted estimator.

**Theorem 3.13** (Convergence rates of pre-training-reweighted estimator). *Suppose Assumptions 1 to 7 hold. Assume that $P_X$ and $Q_X$ are absolutely continuous with respect to the Lebesgue measure. Let $\mathcal{S} = \{(X_i^\mu, Z_i^\mu)\}_{i=1}^m$ and $\mathcal{D} = \{(X_i^P, Y_i^P)\}_{i=1}^n$ be two i.i.d. sample sets drawn from $\mu$ and $P$, respectively. Set the hypothesis class $\mathcal{U}$ as $\mathcal{U} = \mathcal{N}(W_\mathcal{U}, L_\mathcal{U})$ with width $W_\mathcal{U} = \mathcal{O}(U_\mathcal{U} \log U_\mathcal{U})$ and depth $L_\mathcal{U} = \mathcal{O}(N_\mathcal{U} \log N_\mathcal{U})$ satisfying $U_\mathcal{U} N_\mathcal{U} = \mathcal{O}(n^{\frac{d}{2d+4\alpha}})$. Further, set the hypothesis class $\mathcal{F}$ as $\mathcal{F} = \mathcal{N}(W_\mathcal{F}, L_\mathcal{F})$ with width $W_\mathcal{F} = \mathcal{O}(U \log U)$ and depth $L_\mathcal{F} = \mathcal{O}(N \log N)$ satisfying $U_\mathcal{F} N_\mathcal{F} = \mathcal{O}(\Lambda^{-\frac{d}{2d+4\beta}} n^{\frac{d}{2d+4\beta}})$. Given $\hat{u}_\mathcal{S} \in \mathcal{U}$ defined in (2.9) and $\hat{f}_{\hat{\varrho}_\mathcal{S}, \mathcal{D}} \in \mathcal{F}$ defined in (2.10), then the following inequality holds*

$$\mathbb{E}_{\mathcal{S} \sim \mu^m} \mathbb{E}_{\mathcal{D} \sim P^n} \left[ \|\hat{f}_{\hat{\varrho}_\mathcal{S}, \mathcal{D}} - f_0\|_{L^2(Q_X)}^2 \right] \leq \mathcal{O}\left( \Lambda^{\frac{2\beta}{d+2\beta}} (\log \Lambda) n^{-\frac{2\beta}{d+2\beta}} (\log n)^2 \right) + \mathcal{O}\left( m^{-\frac{\alpha}{d+2\alpha}} \log m \right).$$

*If the pre-training sample size $m$ satisfies $m \geq \mathcal{O}(\Lambda^{-\frac{2\beta}{d+2\beta} \frac{d+2\alpha}{\alpha}} n^{\frac{2\beta}{d+2\beta} \frac{d+2\alpha}{\alpha}})$, then the following inequality holds*

$$\mathbb{E}_{\mathcal{S} \sim \mu^m} \mathbb{E}_{\mathcal{D} \sim P^n} \left[ \|\hat{f}_{\hat{\varrho}_\mathcal{S}, \mathcal{D}} - f_0\|_{L^2(Q_X)}^2 \right] \leq \mathcal{O}\left( \Lambda^{\frac{2\beta}{d+2\beta}} (\log \Lambda) n^{-\frac{2\beta}{d+2\beta}} (\log n)^2 \right).$$

*Remark* 3.14. Theorem 3.13 establishes that, subsequent to the pre-training operation, the resulting pre-training-reweighted estimator can achieve the same nonparametric efficiency as that in Theorem 3.8. This holds true under the condition that the pre-training sample size $m$ satisfies $m \geq \mathcal{O}(\Lambda^{-\frac{2\beta}{d+2\beta} \frac{d+2\alpha}{\alpha}} n^{\frac{2\beta}{d+2\beta} \frac{d+2\alpha}{\alpha}})$, which provides valuable guidance for selecting an appropriate pre-training sample size. Of paramount significance is the observation that this condition is often straightforward to fulfill in practical applications. This is attributable to the fact that collecting unlabeled data is typically more cost-effective than acquiring labeled data in many practical seniors, rendering it a more feasible operation.

## 4 CONCLUSION

This study investigates nonparametric regression under covariate shift and then introduces a two-stage pre-training reweighted approach. We focus on three estimators based on deep ReLU neural networks: the unweighted estimator, reweighted estimator, and pre-training reweighted estimator. We establish rigorous convergence rates for these estimators, wherein our technical novelty lies in using local and offset complexity techniques for statistical error analysis, resulting in a fast rate of $\mathcal{O}(1/n)$. These theoretical results shed light on the significance of density-ratio reweighting strategy and offer a priori guide for selecting the appropriate number of pre-training samples.

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

## A  RELATED TOPICS

In this section, we discuss several research topics related to the covariate shift.

**Out-of-Distribution Generalization.** The out-of-distribution (OOD) generalization problem, as elucidated by Shen et al. (2021), represents a specific facet of the supervised learning paradigm wherein the test distribution, denoted as $Q$, exhibits a notable divergence from the training distribution, symbolized as $P$. It is imperative to emphasize that the test distribution remains unknown during the entirety of the training phase. Within the framework of OOD generalization, the observed distribution shift can be categorized into two principal types, namely concept shifts (Gama et al., 2014; Cai & Wei, 2021) and covariate shifts (Shimodaira, 2000). Concept shifts manifest as alterations in the conditional distribution, denoted as $P_{Y|X}$, resulting in a mismatch with the test distribution $Q_{Y|X}$. Conversely, covariate shifts are characterized by perturbations in the marginal distribution $P_X$, leading to incongruence with the test distribution $Q_X$.

**Transfer Learning and Domain Adaptation.** Transductive transfer learning constitutes a pivotal domain within the broader framework of transfer learning, operating under the invariance assumption that $P_{Y|X} = Q_{Y|X}$. Notably, this assumption has been expounded upon by Pan & Yang (2009); Zhuang et al. (2020). Transductive transfer learning can be dissected into two distinct scenarios, each with its unique characteristics. The first scenario pertains to instances where the feature spaces of the source and target distributions differ, i.e., $\mathcal{X}_P \neq \mathcal{X}_Q$. Conversely, the second scenario involves cases wherein the feature spaces in both the source and target domains remain identical, i.e., $\mathcal{X}_P = \mathcal{X}_Q$, while there exist disparities in the marginal probability distributions of the input data, specifically $P_X \neq Q_X$. This latter case is commonly referred to as domain adaptation, predicated on the assumption that prior knowledge regarding the test distribution is available, encompassing either the joint distribution $Q$ or the marginal distribution $Q_X$. This paper primarily addresses the latter scenario of domain adaptation, wherein the alignment of marginal distributions between the source and target domains becomes the focal point of consideration. Furthermore, it is noteworthy that ratio-reweighting emerges as a crucial mechanism within the domain adaptation framework, as acknowledged by Huang et al. (2006); Belkin et al. (2006); Sugiyama et al. (2008); Sun et al. (2011).

**Stable Learning.** In the context of machine learning, when provided with a training dataset $\{(X_i, Y_i)\}_{i=1}^n$, wherein $\{X_i\}_{i=1}^n$ are sampled from a single distribution defined over the feature space $\mathcal{X}$, the primary objective of stable learning is to formulate an estimator that exhibits consistent and uniformly excellent performance across the entirety of possible distributions over $\mathcal{X}$. To address this formidable challenge, Shen et al. (2020) introduced an important technique known as sample weighting. This technique serves as a pivotal tool in the pursuit of stable learning by assigning appropriate weights to individual data points, thus allowing the learning process to emphasize those instances that contribute significantly to achieving the desired uniformity of performance across diverse distributions over $\mathcal{X}$.

# B  SUPPLEMENTARY DEFINITIONS AND LEMMAS

**Definition B.1** (Hölder's class). Let $\mathcal{X} \subseteq [0,1]^d$ and $\beta = s + r$ with $s \in \mathbb{N}$ and $r \in (0,1]$. The Hölder's class $\mathcal{H}^\beta(\mathcal{X})$ is defined by

$$\mathcal{H}^\beta(\mathcal{X}) = \left\{ f : \mathcal{X} \to \mathbb{R}, \|f\|_{\mathcal{H}^\beta(\mathcal{X})} := \max_{|\alpha| \leq s} \|\partial^\alpha f\|_\infty + \max_{|\alpha|=s} \sup_{x \neq y} \frac{|\partial^\alpha f(x) - \partial^\alpha f(y)|}{\|x-y\|_2^r} < \infty \right\},$$

where $\partial^\alpha = \partial^{\alpha_1} \cdots \partial^{\alpha_d}$ with $\alpha = (\alpha_1, \ldots, \alpha_d)^T$. Moreover, for some positive constant $B$, the bounded Hölder's class is defined by $\mathcal{H}^\beta(\mathcal{X}, B) = \{f : \mathcal{X} \to \mathbb{R}, \|f\|_{\mathcal{H}^\beta(\mathcal{X})} \leq B\}$.

**Definition B.2** (VC-dimension). Let $\mathcal{F}$ be a class of functions from $\mathcal{X}$ to $\{\pm 1\}$. For any non-negative integer $m$, we define the growth function of $\mathcal{F}$ as

$$\Pi_\mathcal{F}(m) = \max_{\{x_i\}_{i=1}^m \subseteq \mathcal{X}} \big| \{(f(x_1), \ldots, f(x_m)) : f \in \mathcal{F}\} \big|.$$

A set $\mathcal{X} = \{x_i\}_{i=1}^m$ is said to be shattered by $\mathcal{F}$ when $|\{(f(x_1), \ldots, f(x_m)) : f \in \mathcal{F}\}| = 2^m$. The Vapnik-Chervonenkis dimension of $\mathcal{F}$, denoted $\mathrm{VCdim}(\mathcal{F})$, is the size of the largest set that can be shattered by $\mathcal{F}$, that is, $\mathrm{VCdim}(\mathcal{F}) = \max\{m : \Pi_\mathcal{F}(m) = 2^m\}$. For a class $\mathcal{F}$ of real-valued functions, we define $\mathrm{VCdim}(\mathcal{F}) = \mathrm{VCdim}(\mathrm{sign}(\mathcal{F}))$.

**Definition B.3** (Covering number). Let $\mathcal{F}$ be a class of measurable functions from $\mathcal{X}$ to $\mathbb{R}$ and $\mathcal{X} = \{X_i\}_{i=1}^m \subseteq \mathcal{X}$. Define the $L^p(\mathcal{X})$-norm as

$$\|f\|_{L^p(\mathcal{X})} = \left( \frac{1}{n} \sum_{i=1}^n |f(X_i)|^p \right)^{1/p}, \quad \text{for } 1 \leq p < \infty,$$

and $\|f\|_{L^\infty(\mathcal{X})} = \max_{1 \leq i \leq m} |f(X_i)|$. A set $\mathcal{F}_\delta$ is called a $L^p(\mathcal{X})$ $\delta$-cover of $\mathcal{F}$ if for each $f \in \mathcal{F}$, there exits $f_\delta \in \mathcal{F}_\delta$ such that $\|f - f_\delta\|_{L^p(\mathcal{X})} \leq \delta$. Furthermore,

$$N(\delta, \mathcal{F}, L^p(\mathcal{X})) = \inf \big\{ |\mathcal{F}_\delta| : \mathcal{F}_\delta \text{ is a } L^p(\mathcal{X}) \ \delta\text{-cover of } \mathcal{F} \big\}$$

is called the empirical $\delta$-covering number of $\mathcal{F}$ based the sample $\mathcal{X} = \{X_i\}_{i=1}^n$.

**Lemma B.4.** *Let $\xi_j$ be a $\sigma^2$-sub-Gaussian random variable for $j \in [N]$. Then*

$$\mathbb{E}\left[ \max_{1 \leq j \leq N} \xi_j^2 \right] \leq 4\sigma^2 (\log N + 1).$$

*Proof of Lemma B.4.* By Jensen's inequality, it is straightforward that

$$\exp\left( \frac{\lambda}{2\sigma^2} \mathbb{E}\left[ \max_{1 \leq j \leq N} \xi_j^2 \right] \right) \leq \mathbb{E}\left[ \max_{1 \leq j \leq N} \exp\left( \frac{\lambda \xi_j^2}{2\sigma^2} \right) \right] \leq N \mathbb{E}\left[ \exp\left( \frac{\lambda \xi_1^2}{2\sigma^2} \right) \right] \leq \frac{N}{\sqrt{1-\lambda}},$$

where the last inequality holds from Wainwright (2019, Theorem 2.6) for each $\lambda \in [0,1)$. Letting $\lambda = 1/2$ yields the desired inequality. $\square$

*Proof of Lemma 2.1.* Setting the first variation of

$$\int_\mathcal{X} \log(1 + \exp(-u(x))) p_X(x) + \log(1 + \exp(u(x))) q_X(x) dx$$

to zero yields

$$\frac{\exp(-u(x))}{1 + \exp(-u(x))} p_X(x) = \frac{\exp(u(x))}{1 + \exp(u(x))} q_X(x),$$

which completes the proof. $\square$

## C    ERROR ANALYSIS FOR UNWEIGHTED ESTIMATORS

*Proof of Lemma 3.3.* According to Assumption 2, it is easy to show that

$$\|\hat{f}_{\mathcal{D}} - f_0\|^2_{L^2(Q_X)} = \mathbb{E}_{X^P \sim P_X}\big[(\hat{f}_{\mathcal{D}}(X^P) - f_0(X^P))^2 \varrho(X^P)\big] \leq \Lambda \|\hat{f}_{\mathcal{D}} - f_0\|^2_{L^2(P_X)}.$$

Define $R(f) = \mathbb{E}_{(X^P, Y^P) \sim P}[(f(X^P) - Y^P)^2]$. By setting $\varrho(x) \equiv 1$ in the Step 1 of the proof of Lemma 3.7, we find that

$$\mathbb{E}_{\mathcal{D}}\big[R(\hat{f}_{\mathcal{D}}) - 3\widehat{R}_{\mathcal{D}}(\hat{f}_{\mathcal{D}})\big] \leq 24B^2 \frac{\text{VCdim}(\mathcal{F})\log(en)}{n}, \tag{C.1}$$

where $\widehat{R}_{\mathcal{D}}(f) := \frac{1}{n}\sum_{i=1}^n (f(X_i^P) - f_0(X_i^P))^2$, $f \in \mathcal{F}$. Similarly, it holds from the Step 2 in the proof of Lemma 3.7 that

$$\mathbb{E}_{\mathcal{D}}\Big[\widehat{R}_{\mathcal{D}}(\hat{f}_{\mathcal{D}})\Big] \leq 2\inf_{f \in \mathcal{F}}\|f - f_0\|^2_{L^2(P_X)} + 100\sigma^2 \frac{\text{VCdim}(\mathcal{F})\log(en)}{n} + \frac{4B^2}{n}. \tag{C.2}$$

Combining (C.1) and (C.2) completes the proof. $\qquad\square$

*Proof of Theorem 3.4.* According to Assumption 4 and Lemma 3.1, there exists $f \in \mathcal{F} = N(W_{\mathcal{F}}, L_{\mathcal{F}})$ such that

$$\inf_{f \in \mathcal{F}}\|f - f_0\|^2_{L^2(P_X)} \leq C_1 (U_{\mathcal{F}}N_{\mathcal{F}})^{-4\beta/d},$$

where $W = \mathcal{O}(U_{\mathcal{F}}\log U_{\mathcal{F}})$, $L = \mathcal{O}(N_{\mathcal{F}}\log N_{\mathcal{F}})$ and $C_1$ is a constant only depending on $\|f_0\|_{\mathcal{H}^\beta(\mathcal{X})}$, $d$ and $\beta$. Using Lemma 3.2, we find that $\text{VCdim}(\mathcal{F}) \leq C_2 U_{\mathcal{F}}^2 N_{\mathcal{F}}^2 (\log U_{\mathcal{F}}\log N_{\mathcal{F}})^2$, where the constant $C_2$ depends on $B$, $d$ and $\beta$. By setting $U_{\mathcal{F}}N_{\mathcal{F}} = \mathcal{O}(n^{\frac{d}{2d+4\beta}})$, we conclude the final result. $\qquad\square$

## D    ERROR ANALYSIS FOR REWEIGHTED ESTIMATORS

Recall that $\mathcal{D} = \{(X_i^P, Y_i^P)\}_{i=1}^n$ are i.i.d. drawn from the probability distribution $P$. We define the reweighted excess risk $R_\varrho$ by

$$R_\varrho(f) = \mathbb{E}_{X^P \sim P_X}\big[\varrho(X^P)(f(X^P) - f_0(X^P))^2\big],$$

and its empirical counterpart $\widehat{R}_{\varrho,\mathcal{D}}$ based on $\mathcal{D}$ can be given by

$$\widehat{R}_{\varrho,\mathcal{D}}(f) = \frac{1}{n}\sum_{i=1}^n \varrho(X_i^P)(f(X_i^P) - f_0(X_i^P))^2.$$

It is easy to verify that $R_\varrho(f) = \|f - f_0\|^2_{L^2(Q_X)}$.

*Proof of Lemma 3.7.* To begin with, it is straightforward that

$$\mathbb{E}_{\mathcal{D}}\Big[\|\hat{f}_{\varrho,\mathcal{D}} - f_0\|^2_{L^2(Q_X)}\Big] = \mathbb{E}_{\mathcal{D}}\Big[R_\varrho(\hat{f}_{\varrho,\mathcal{D}}) - 3\widehat{R}_{\varrho,\mathcal{D}}(\hat{f}_{\varrho,\mathcal{D}})\Big] + 3\mathbb{E}_{\mathcal{D}}\Big[\widehat{R}_{\varrho,\mathcal{D}}(\hat{f}_{\varrho,\mathcal{D}})\Big]. \tag{D.1}$$

We now derive the upper bound of these two parts on the right hand of (D.1), respectively.

*Step 1. Symmetrization by a ghost sample.*

We define the function class $\mathcal{H} = \{x \mapsto h(x) = \varrho(x)(f(x) - f_0(x))^2 : f \in \mathcal{F}\}$. Since that $0 \leq \varrho(x) \leq \Lambda$ for each $x \in \mathcal{X}$, it is apparent that $0 \leq h(x) \leq 4\Lambda B^2$ for each $x \in \mathcal{X}$ and $h \in \mathcal{H}$. Then it is easy to show that

$$\mathbb{E}_{\mathcal{D}}\big[R_\varrho(\hat{f}_{\varrho,\mathcal{D}}) - 3\widehat{R}_{\varrho,\mathcal{D}}(\hat{f}_{\varrho,\mathcal{D}})\big] \leq \mathbb{E}_{\mathcal{D}}\sup_{h \in \mathcal{H}}\Big\{\mathbb{E}_{X^P}[h(X^P)] - \frac{3}{n}\sum_{i=1}^n h(X_i^P)\Big\}$$

$$\leq \mathbb{E}_{\mathcal{D}}\sup_{h \in \mathcal{H}}\Big\{2\mathbb{E}_{X^P}[h(X^P)] - \frac{1}{4\Lambda B^2}\mathbb{E}_{X^P}[h^2(X^P)] - \frac{2}{n}\sum_{i=1}^n h(X_i^P) - \frac{1}{4\Lambda B^2 n}\sum_{i=1}^n h^2(X_i^P)\Big\},$$

where we used the fact that $h^2(x) \leq 4\Lambda B^2 h(x)$ for each $x \in \mathcal{X}$ and $h \in \mathcal{H}$.

Let us introduce a ghost sample $\mathcal{D}' = \{(X_i^{P,\prime}, Y_i^{P,\prime})\}_{i=1}^n$ sampled from $P$, which is independent of $\mathcal{D}$. Let $\{\varepsilon_i\}_{i=1}^n$ be a set of Rademacher variables. Then replacing the expectation by the empirical mean based on the ghost sample $\mathcal{D}'$ yields

$$\mathbb{E}_{\mathcal{D}} \sup_{h \in \mathcal{H}} \left\{ 2\mathbb{E}_{X^P}[h(X^P)] - \frac{1}{4\Lambda B^2}\mathbb{E}_{X^P}[h^2(X^P)] - \frac{2}{n}\sum_{i=1}^n h(X_i^P) - \frac{1}{4\Lambda B^2 n}\sum_{i=1}^n h^2(X_i^P) \right\}$$

$$= \mathbb{E}_{\mathcal{D}} \sup_{h \in \mathcal{H}} \left\{ 2\mathbb{E}_{\mathcal{D}'}\left[\frac{1}{n}\sum_{i=1}^n (h(X_i^{P,\prime}) - h(X_i^P))\right] - \frac{1}{4\Lambda B^2}\mathbb{E}_{\mathcal{D}'}\left[\frac{1}{n}\sum_{i=1}^n (h^2(X_i^{P,\prime}) + h^2(X_i^P))\right] \right\}$$

$$\leq \mathbb{E}_{\mathcal{D}}\mathbb{E}_{\mathcal{D}'} \sup_{h \in \mathcal{H}} \left\{ \frac{2}{n}\sum_{i=1}^n (h(X_i^{P,\prime}) - h(X_i^P)) - \frac{1}{4\Lambda B^2}\frac{1}{n}\sum_{i=1}^n (h^2(X_i^{P,\prime}) + h^2(X_i^P)) \right\}$$

$$= \mathbb{E}_{\mathcal{D}}\mathbb{E}_{\mathcal{D}'}\mathbb{E}_{\varepsilon} \sup_{h \in \mathcal{H}} \left\{ \frac{2}{n}\sum_{i=1}^n \varepsilon_i(h(X_i^{P,\prime}) - h(X_i^P)) - \frac{1}{4\Lambda B^2}\frac{1}{n}\sum_{i=1}^n (h^2(X_i^{P,\prime}) + h^2(X_i^P)) \right\}$$

$$= \mathbb{E}_{\mathcal{D}}\mathbb{E}_{\varepsilon} \sup_{h \in \mathcal{H}} \left\{ \frac{2}{n}\sum_{i=1}^n \varepsilon_i h(X_i^P) - \frac{1}{4\Lambda B^2}\frac{1}{n}\sum_{i=1}^n h^2(X_i^P) \right\},$$

where $\mathbb{E}_{\varepsilon}[\cdot]$ is the expectation conditional on $\mathcal{D}$ and $\mathcal{D}'$, and the inequality holds from Jensen's inequality. We note that $\mathbb{E}_{\mathcal{D}}\mathbb{E}_{\varepsilon} \sup_{h \in \mathcal{H}} \left\{ \frac{2}{n}\sum_{i=1}^n \varepsilon_i h(X_i^P) - \frac{1}{4\Lambda B^2}\frac{1}{n}\sum_{i=1}^n h^2(X_i^P) \right\}$ refers to the offset Rademacher complexity (Liang et al., 2015). Then we transform into bounding this offset Rademacher complexity to derive the upper bound.

Let $\delta \in (0, 4\Lambda B^2)$ and $\mathcal{H}_\delta$ be a $L^\infty(\mathcal{D})$ $\delta$-cover of $\mathcal{H}$ satisfying $|N_\delta| = N(\delta, \mathcal{H}, L^\infty(\mathcal{D}))$. Then for each $h \in \mathcal{H}$, there exists $h_\delta \in \mathcal{H}_\delta$ such that $\max_{1 \leq i \leq n} |h(X_i^P) - h_\delta(X_i^P)| \leq \delta$. Consequently, it follows from Hölder's inequality that

$$\frac{1}{n}\sum_{i=1}^n \varepsilon_i h(X_i^P) \leq \frac{1}{n}\sum_{i=1}^n \varepsilon_i h_\delta(X_i^P) + \frac{1}{n}\sum_{i=1}^n |\varepsilon_i||h(X_i^P) - h_\delta(X_i^P)| \leq \frac{1}{n}\sum_{i=1}^n \varepsilon_i h_\delta(X_i^P) + \delta,$$

and

$$-\frac{1}{n}\sum_{i=1}^n h^2(X_i^P) \leq -\frac{1}{n}\sum_{i=1}^n h_\delta^2(X_i^P) + \frac{1}{n}\sum_{i=1}^n |h(X_i^P) + h_\delta(X_i^P)||h_\delta(X_i^P) - h(X_i^P)|$$

$$\leq -\frac{1}{n}\sum_{i=1}^n h_\delta^2(X_i^P) + 8\Lambda B^2\delta.$$

Hence we find that

$$\mathbb{E}_{\varepsilon} \sup_{h \in \mathcal{H}} \left\{ \frac{2}{n}\sum_{i=1}^n \varepsilon_i h(X_i^P) - \frac{1}{4\Lambda B^2 n}\sum_{i=1}^n h^2(X_i^P) \right\}$$

$$\leq \mathbb{E}_{\varepsilon} \max_{h_\delta \in \mathcal{H}_\delta} \left\{ \frac{2}{n}\sum_{i=1}^n \varepsilon_i h_\delta(X_i^P) - \frac{1}{4\Lambda B^2 n}\sum_{i=1}^n h_\delta^2(X_i^P) \right\} + 4\delta. \tag{D.2}$$

By Hoeffding's inequality (Mohri et al., 2018, Theorem D.2), the conditional probability can be bounded as follows

$$\Pr\left( \frac{2}{n}\sum_{i=1}^n \varepsilon_i h_\delta(X_i^P) > t + \frac{1}{4\Lambda B^2}\frac{1}{n}\sum_{i=1}^n h_\delta^2(X_i^P) \Big| \mathcal{D} = \{(X_i^P, Y_i^P)\}_{i=1}^n \right)$$

$$= \Pr\left( \sum_{i=1}^n \varepsilon_i h_\delta(X_i^P) > \frac{nt}{2} + \frac{1}{8\Lambda B^2}\sum_{i=1}^n h_\delta^2(X_i^P) \Big| \mathcal{D} = \{(X_i^P, Y_i^P)\}_{i=1}^n \right)$$

$$\leq \exp\left( -\frac{(\frac{nt}{2} + \frac{1}{8\Lambda B^2}\sum_{i=1}^n h_\delta^2(X_i^P))^2}{2\sum_{i=1}^n h_\delta^2(X_i^P)} \right) \leq \exp\left( -\frac{nt}{8\Lambda B^2} \right).$$

As a consequence, it follows that for each $T > 0$,

$$
\mathbb{E}_\varepsilon \max_{h_\delta \in \mathcal{H}_\delta} \Big\{ \frac{2}{n} \sum_{i=1}^n \varepsilon_i h_\delta(X_i^P) - \frac{1}{4\Lambda B^2 n} \sum_{i=1}^n h_\delta^2(X_i^P) \Big\}
$$

$$
\leq \int_0^\infty \Pr \Big( \max_{h_\delta \in \mathcal{H}_\delta} \Big\{ \frac{2}{n} \sum_{i=1}^n \varepsilon_i h_\delta(X_i^P) - \frac{1}{4\Lambda B^2 n} \sum_{i=1}^n h_\delta^2(X_i^P) \Big\} > t \Big| \mathcal{D} = \{(X_i^P, Y_i^P)\}_{i=1}^n \Big) dt
$$

$$
\leq T + |\mathcal{H}_\delta| \int_T^\infty \Pr \Big( \frac{2}{n} \sum_{i=1}^n \varepsilon_i h_\delta(X_i^P) > t + \frac{1}{4\Lambda B^2 n} \sum_{i=1}^n h_\delta^2(X_i^P) \Big| \mathcal{D} = \{(X_i^P, Y_i^P)\}_{i=1}^n \Big) dt
$$

$$
\leq T + \frac{8\Lambda B^2}{n} |\mathcal{H}_\delta| \exp \Big( - \frac{nT}{8\Lambda B^2} \Big).
$$

Letting $T = \frac{8\Lambda B^2}{n} \log |\mathcal{H}_\delta|$ gives that

$$
\mathbb{E}_\varepsilon \max_{h_\delta \in \mathcal{H}_\delta} \Big\{ \frac{2}{n} \sum_{i=1}^n \varepsilon_i h_\delta(X_i^P) - \frac{1}{4\Lambda B^2 n} \sum_{i=1}^n h_\delta^2(X_i^P) \Big\} \leq \frac{8\Lambda B^2}{n} (\log |\mathcal{H}_\delta| + 1). \tag{D.3}
$$

It remains to estimate the covering number $|\mathcal{H}_\delta|$. Noticing

$$
|h(x) - h'(x)| = |\varrho(x)| |(f(x) - f_0(x))^2 - (f'(x) - f_0(x))^2| \leq 4\Lambda B |f(x) - f'(x)|,
$$

we find that

$$
\log N(\delta, \mathcal{H}, L^\infty(\mathcal{D})) \leq \log N \Big( \frac{\delta}{4\Lambda B}, \mathcal{F}, L^\infty(\mathcal{D}) \Big) \leq \mathrm{VCdim}(\mathcal{F}) \log \Big( \frac{4e\Lambda B^2 n}{\delta} \Big), \tag{D.4}
$$

where the last inequality is owing to Anthony et al. (1999, Theorem 12.2). By setting $\delta = 4\Lambda B^2/n$, it holds from (D.2) to (D.4) that

$$
\mathbb{E}_\mathcal{D} \big[ R_\varrho(\hat{f}_{\varrho,\mathcal{D}}) - 3\widehat{R}_{\varrho,\mathcal{D}}(\hat{f}_{\varrho,\mathcal{D}}) \big] \leq 24\Lambda B^2 \frac{\mathrm{VCdim}(\mathcal{F}) \log(en)}{n}. \tag{D.5}
$$

*Step 2. Estimate of empirical excess risk.*

For each function $f : \mathcal{X} \to \mathbb{R}$, we connect the empirical risk of it with its empirical excess risk by

$$
\widehat{R}_{\varrho,\mathcal{D}}(f) = \widehat{L}_{\varrho,\mathcal{D}}(f) + \frac{2}{n} \sum_{i=1}^n \varrho(X_i^P) \xi_i (f(X_i^P) - f_0(X_i^P)) - \frac{1}{n} \sum_{i=1}^n \varrho(X_i^P) \xi_i^2.
$$

Plugging the reweighted empirical risk minimizer $\hat{f}_{\varrho,\mathcal{D}}$ and taking expectation with respect to $\mathcal{D}$ on both sides of the equality implies that for each $f \in \mathcal{F}$,

$$
\mathbb{E}_\mathcal{D} \big[ \widehat{R}_{\varrho,\mathcal{D}}(\hat{f}_{\varrho,\mathcal{D}}) \big] = \mathbb{E}_\mathcal{D} \big[ \widehat{L}_{\varrho,\mathcal{D}}(\hat{f}_{\varrho,\mathcal{D}}) \big] - \sigma^2 + 2\mathbb{E}_\mathcal{D} \Big[ \frac{1}{n} \sum_{i=1}^n \varrho(X_i^P) \xi_i \hat{f}_{\varrho,\mathcal{D}}(X_i^P) \Big]
$$

$$
\leq \mathbb{E}_\mathcal{D} \big[ \widehat{L}_{\varrho,\mathcal{D}}(f) \big] - \sigma^2 + 2\mathbb{E}_\mathcal{D} \Big[ \frac{1}{n} \sum_{i=1}^n \varrho(X_i^P) \xi_i \hat{f}_{\varrho,\mathcal{D}}(X_i^P) \Big]
$$

$$
= \|f - f_0\|_{L^2(Q_X)}^2 + 2\mathbb{E}_\mathcal{D} \Big[ \frac{1}{n} \sum_{i=1}^n \varrho(X_i^P) \xi_i \hat{f}_{\varrho,\mathcal{D}}(X_i^P) \Big],
$$

which deduces

$$
\mathbb{E}_\mathcal{D} \big[ \widehat{R}_{\varrho,\mathcal{D}}(\hat{f}_{\varrho,\mathcal{D}}) \big] \leq \inf_{f \in \mathcal{F}} \|f - f_0\|_{L^2(Q_X)}^2 + 2\mathbb{E}_\mathcal{D} \Big[ \frac{1}{n} \sum_{i=1}^n \varrho(X_i^P) \xi_i \hat{f}_{\varrho,\mathcal{D}}(X_i^P) \Big]. \tag{D.6}
$$

Define $\hat{g}_\mathcal{D}(x) = \varrho(x) \hat{f}_{\varrho,\mathcal{D}}(x)$ and $g_0 = \varrho(x) f_0(x)$ for each $x \in \mathcal{X}$. In addition, define the function class $\mathcal{G} = \{x \mapsto g(x) = \varrho(x) f(x) : f \in \mathcal{F}\}$. Let $\delta \in (0, \Lambda B)$ and $\mathcal{G}_\delta$ be a $L^\infty(\mathcal{D})$ $\delta$-cover of $\mathcal{G}$

with $|\mathcal{G}_\delta| = N(\delta, \mathcal{G}, L^\infty(\mathcal{D}))$. Suppose that $g_\delta$ is a function in $\mathcal{G}_\delta$ such that $\max_{1 \le i \le n} |\hat{g}_\mathcal{D}(X_i^P) - g_\delta(X_i^P)| \le \delta$. Then we find that

$$\mathbb{E}_\mathcal{D}\Big[\frac{1}{n}\sum_{i=1}^n \xi_i(\hat{g}_\mathcal{D}(X_i^P) - g_\delta(X_i^P))\Big] \le \delta\mathbb{E}_\mathcal{D}\Big[\frac{1}{n}\sum_{i=1}^n |\xi_i|\Big] \le \delta\sigma,$$

where the last inequality is due to Hölder's inequality. Consequently, we have

$$\mathbb{E}_\mathcal{D}\Big[\frac{1}{n}\sum_{i=1}^n \xi_i\hat{g}_\mathcal{D}(X_i^P)\Big] = \mathbb{E}_\mathcal{D}\Big[\frac{1}{n}\sum_{i=1}^n \xi_i(\hat{g}_\mathcal{D}(X_i^P) - g_0(X_i^P))\Big]$$

$$\le \mathbb{E}_\mathcal{D}\Big[\frac{1}{n}\sum_{i=1}^n \xi_i(g_\delta(X_i^P) - g_0(X_i^P))\Big] + \delta\sigma$$

$$\le \mathbb{E}_\mathcal{D}\Big[\frac{\|\hat{g}_\mathcal{D} - g_0\|_{L^2(\mathcal{D})} + \delta}{\sqrt{n}}\psi(g_\delta)\Big] + \delta\sigma$$

$$\le \Big(\mathbb{E}_\mathcal{D}^{1/2}\Big[\|\hat{g}_\mathcal{D} - g_0\|_{L^2(\mathcal{D})}^2\Big] + \delta\Big)\frac{1}{\sqrt{n}}\mathbb{E}_\mathcal{D}^{1/2}\Big[\psi^2(g_\delta)\Big] + \delta\sigma$$

$$\le \frac{1}{4}\mathbb{E}_\mathcal{D}\Big[\|\hat{g}_\mathcal{D} - g_0\|_{L^2(\mathcal{D})}^2\Big] + \frac{2}{n}\mathbb{E}_\mathcal{D}\Big[\psi^2(g_\delta)\Big] + \frac{1}{4}\delta^2 + \delta\sigma. \quad \text{(D.7)}$$

Here, the first and second inequalities are from the definition of covering, and

$$\psi(g_\delta) := \frac{\sum_{i=1}^n \xi_i(g_\delta(X_i^P) - g_0(X_i^P))}{\sqrt{n}\|g_\delta - g_0\|_{L^2(\mathcal{D})}},$$

the third inequality holds from Cauchy-Schwarz inequality, while the last one is owing to the AM-GM inequality $ab \le a^2/4 + b^2$. Observe that for each fixed $g_\delta$, the random variable $\psi(g_\delta)$ is sub-Gaussian with variance proxy $\sigma^2$. Then using Lemma B.4 gives that

$$\mathbb{E}_\xi\Big[\psi^2(g_\delta)\Big] \le \mathbb{E}_\xi\Big[\max_{g \in \mathcal{G}_\delta}\psi^2(g)\Big] \le 4\sigma^2(\log|\mathcal{G}_\delta| + 1). \quad \text{(D.8)}$$

We now estimate the covering number $|\mathcal{G}_\delta|$. Using the fact that

$$|g(x) - g'(x)| = |\varrho(x)||f(x) - f'(x)| \le \Lambda|f(x) - f'(x)|,$$

we implies for $n \ge \text{VCdim}(\mathcal{F})$,

$$\log N(\delta, \mathcal{G}, L^\infty(\mathcal{D})) \le \log N\Big(\frac{\delta}{\Lambda}, \mathcal{F}, L^\infty(\mathcal{D})\Big) \le \text{VCdim}(\mathcal{F})\log\Big(\frac{e\Lambda Bn}{\delta}\Big), \quad \text{(D.9)}$$

where the last inequality is due to Anthony et al. (1999, Theorem 12.2). Combining (D.7) to (D.9) and setting $\delta = B/n$ gives

$$\mathbb{E}_\mathcal{D}\Big[\frac{1}{n}\sum_{i=1}^n \xi_i\hat{g}_\mathcal{D}(X_i^P)\Big] \le \frac{1}{4}\mathbb{E}_\mathcal{D}\Big[\widehat{R}_{\varrho,\mathcal{D}}(\hat{f}_{\varrho,\mathcal{D}})\Big] + 25\sigma^2\frac{\text{VCdim}(\mathcal{F})\log(e\Lambda n)}{n} + \frac{B^2}{n}. \quad \text{(D.10)}$$

Using (D.6) and (D.10) yields

$$\mathbb{E}_\mathcal{D}\Big[\widehat{R}_{\varrho,\mathcal{D}}(\hat{f}_{\varrho,\mathcal{D}})\Big] \le 2\inf_{f \in \mathcal{F}}\|f - f_0\|_{L^2(Q_X)}^2 + 100\sigma^2\frac{\text{VCdim}(\mathcal{F})\log(e\Lambda n)}{n} + \frac{4B^2}{n}. \quad \text{(D.11)}$$

Combining (D.1), (D.5) and (D.11) completes the proof. □

*Proof of Theorem 3.8.* According to Assumption 4 and Lemma 3.1, there exists $f \in \mathcal{F} = N(W_\mathcal{F}, L_\mathcal{F})$ such that

$$\inf_{f \in \mathcal{F}}\|f - f_0\|_{L^2(Q_X)}^2 \le C_1(U_\mathcal{F}N_\mathcal{F})^{-4\beta/d},$$

where $W = \mathcal{O}(U_\mathcal{F}\log U_\mathcal{F})$, $L = \mathcal{O}(N_\mathcal{F}\log N_\mathcal{F})$ and $C_1$ is a constant only depending on $\|f_0\|_{\mathcal{H}^\beta(\mathcal{X})}$, $d$ and $\beta$. Using Lemma 3.2, we find that $\text{VCdim}(\mathcal{F}) \le C_2U_\mathcal{F}^2N_\mathcal{F}^2(\log U_\mathcal{F}\log N_\mathcal{F})^2$, where the constant $C_2$ depends on $B$, $d$ and $\beta$. By setting $U_\mathcal{F}N_\mathcal{F} = \mathcal{O}(\Lambda^{-\frac{d}{2d+4\beta}}n^{\frac{d}{2d+4\beta}})$, we conclude the final result. □

*Proof of Lemma 3.10.* It is straightforward to verify that

$$
\mathbb{E}_{\mathcal{D}}\Big[\|\hat{f}_{\hat{\varrho}_\mathbb{S},\mathcal{D}} - f_0\|_{L^2(Q_X)}^2\Big] = \mathbb{E}_{\mathcal{D}}\Big[R_\varrho(\hat{f}_{\hat{\varrho}_\mathbb{S},\mathcal{D}}) - R_{\hat{\varrho}_\mathbb{S}}(\hat{f}_{\hat{\varrho}_\mathbb{S},\mathcal{D}})\Big]
$$
$$
+ \mathbb{E}_{\mathcal{D}}\Big[R_{\hat{\varrho}_\mathbb{S}}(\hat{f}_{\hat{\varrho}_\mathbb{S},\mathcal{D}}) - 3\widehat{R}_{\hat{\varrho}_\mathbb{S},\mathcal{D}}(\hat{f}_{\hat{\varrho}_\mathbb{S},\mathcal{D}})\Big] \tag{D.12}
$$
$$
+ 3\mathbb{E}_{\mathcal{D}}\Big[\widehat{R}_{\hat{\varrho}_\mathbb{S},\mathcal{D}}(\hat{f}_{\hat{\varrho}_\mathbb{S},\mathcal{D}})\Big].
$$

Notice that for each $f : \mathcal{X} \to [-B, B]$, it holds that

$$
R_\varrho(f) - R_{\hat{\varrho}_\mathbb{S}}(f) = \mathbb{E}_{X^P \sim P_X}\Big[(\varrho(X^P) - \hat{\varrho}_\mathbb{S}(X^P))(f(X^P) - f_0(X^P))^2\Big]
$$
$$
\leq \mathbb{E}_{X^P \sim P_X}^{1/2}\Big[(\varrho(X^P) - \hat{\varrho}_\mathbb{S}(X^P))^2\Big]\mathbb{E}_{X^P \sim P_X}^{1/2}\Big[(f(X^P) - f_0(X^P))^4\Big]
$$
$$
\leq 4B^2\|\varrho - \hat{\varrho}_\mathbb{S}\|_{L^2(P_X)}, \tag{D.13}
$$

where the first inequality is due to Cauchy-Schwarz inequality. As a consequence, we have

$$
\mathbb{E}_{\mathcal{D}}\Big[R_\varrho(\hat{f}_{\hat{\varrho}_\mathbb{S},\mathcal{D}}) - R_{\hat{\varrho}_\mathbb{S}}(\hat{f}_{\hat{\varrho}_\mathbb{S},\mathcal{D}})\Big] \leq 4B^2\|\varrho - \hat{\varrho}_\mathbb{S}\|_{L^2(P_X)}. \tag{D.14}
$$

We next consider the second and third terms in (D.12). Using a same technique as that used in the Step 3 of the proof of Lemma 3.7, it follows from Assumption 7 that

$$
\mathbb{E}_{\mathcal{D}}\Big[R_{\hat{\varrho}_\mathbb{S}}(\hat{f}_{\hat{\varrho}_\mathbb{S},\mathcal{D}}) - 3\widehat{R}_{\hat{\varrho}_\mathbb{S},\mathcal{D}}(\hat{f}_{\hat{\varrho}_\mathbb{S},\mathcal{D}})\Big] \leq 24\Lambda B^2 \frac{\mathrm{VCdim}(\mathcal{F})\log(en)}{n}. \tag{D.15}
$$

Finally, we estimate the last term in (D.12). To this end, we next relate the reweighted population loss with the reweighted $L^2$-risk by

$$
L_{\hat{\varrho}_\mathbb{S}}(f) = \mathbb{E}_{(X^P, Y^P) \sim P}\Big[\hat{\varrho}_\mathbb{S}(X^P)(f(X^P) - f_0(X^P) - \xi)^2\Big]
$$
$$
= R_{\hat{\varrho}_\mathbb{S}}(f) + \mathbb{E}_{X^P \sim P_X}\Big[\hat{\varrho}_\mathbb{S}(X^P)\Big]\sigma^2. \tag{D.16}
$$

Similarly, their empirical counterparts are related by

$$
\widehat{L}_{\hat{\varrho}_\mathbb{S}}(f) = \widehat{R}_{\hat{\varrho}_\mathbb{S}}(f) - \frac{2}{n}\sum_{i=1}^n \hat{\varrho}_\mathbb{S}(X_i^P)\xi_i(f(X_i^P) - f_0(X_i^P)) + \frac{1}{n}\sum_{i=1}^n \hat{\varrho}_\mathbb{S}(X_i^P)\xi_i^2. \tag{D.17}
$$

Then it follows for each $f \in \mathcal{F}$ that

$$
\mathbb{E}_{\mathcal{D}}\Big[\widehat{R}_{\hat{\varrho}_\mathbb{S},\mathcal{D}}(\hat{f}_{\hat{\varrho}_\mathbb{S},\mathcal{D}})\Big]
$$
$$
= \mathbb{E}_{\mathcal{D}}\Big[\widehat{L}_{\hat{\varrho}_\mathbb{S},\mathcal{D}}(\hat{f}_{\hat{\varrho}_\mathbb{S},\mathcal{D}})\Big] + 2\mathbb{E}_{\mathcal{D}}\Big[\frac{1}{n}\sum_{i=1}^n \hat{\varrho}_\mathbb{S}(X_i^P)\xi_i\hat{f}_{\hat{\varrho}_\mathbb{S},\mathcal{D}}(X_i^P)\Big] - \mathbb{E}_{X^P \sim P_X}\Big[\hat{\varrho}_\mathbb{S}(X^P)\Big]\sigma^2
$$
$$
\leq \mathbb{E}_{\mathcal{D}}\Big[\widehat{L}_{\hat{\varrho}_\mathbb{S},\mathcal{D}}(f)\Big] + 2\mathbb{E}_{\mathcal{D}}\Big[\frac{1}{n}\sum_{i=1}^n \hat{\varrho}_\mathbb{S}(X_i^P)\xi_i\hat{f}_{\hat{\varrho}_\mathbb{S},\mathcal{D}}(X_i^P)\Big] - \mathbb{E}_{X^P \sim P_X}\Big[\hat{\varrho}_\mathbb{S}(X^P)\Big]\sigma^2
$$
$$
= L_{\hat{\varrho}_\mathbb{S}}(f) + 2\mathbb{E}_{\mathcal{D}}\Big[\frac{1}{n}\sum_{i=1}^n \hat{\varrho}_\mathbb{S}(X_i^P)\xi_i\hat{f}_{\hat{\varrho}_\mathbb{S},\mathcal{D}}(X_i^P)\Big] - \mathbb{E}_{X^P \sim P_X}\Big[\hat{\varrho}_\mathbb{S}(X^P)\Big]\sigma^2
$$
$$
= \Big(R_{\hat{\varrho}_\mathbb{S}}(f) - R_\varrho(f)\Big) + R_\varrho(f) + 2\mathbb{E}_{\mathcal{D}}\Big[\frac{1}{n}\sum_{i=1}^n \hat{\varrho}_\mathbb{S}(X_i^P)\xi_i\hat{f}_{\hat{\varrho}_\mathbb{S},\mathcal{D}}(X_i^P)\Big]
$$
$$
\leq 4B^2\|\hat{\varrho}_\mathbb{S} - \varrho\|_{L^2(P_X)} + \|f - f_0\|_{L^2(Q_X)}^2 + 2\mathbb{E}_{\mathcal{D}}\Big[\frac{1}{n}\sum_{i=1}^n \hat{\varrho}_\mathbb{S}(X_i^P)\xi_i\hat{f}_{\hat{\varrho}_\mathbb{S},\mathcal{D}}(X_i^P)\Big],
$$

where the first equality holds from (D.17) and the fact that $\mathbb{E}_\xi[\sum_{i=1}^n \hat{\varrho}_\mathbb{S}(X_i^P)\xi_i f_0(X_i^P)] = 0$, the first inequality holds since $\hat{f}_{\hat{\varrho}_\mathbb{S},\mathcal{D}}$ is the minimizer of $\widehat{L}_{\hat{\varrho}_\mathbb{S}}(f)$ over $\mathcal{F}$, the third equality is due to

(D.16), and the last inequality is from (D.13). Using a similar argument as (D.11) in the Step 3 of the proof of Lemma 3.7, we have

$$
\mathbb{E}_{\mathcal{D}}\left[\widehat{R}_{\hat{\varrho}_{\mathbb{S}},\mathcal{D}}(\hat{f}_{\hat{\varrho}_{\mathbb{S}},\mathcal{D}})\right] \leq 8B^2 \|\hat{\varrho}_{\mathbb{S}} - \varrho\|_{L^2(P_X)} + 2 \inf_{f\in\mathcal{F}} \|f - f_0\|^2_{L^2(Q_X)}
$$
$$
+ 100\sigma^2 \frac{\mathrm{VCdim}(\mathcal{F})\log(e\Lambda n)}{n} + \frac{4B^2}{n}. \tag{D.18}
$$

Combining (D.12), (D.14), (D.15) and (D.18) completes the proof. $\qquad\square$

*Proof of Theorem 3.13.* Combing Lemmas 3.10 and 3.11 and the proof of Theorem 3.8 concludes the final result. $\qquad\square$

# E    ERROR ANALYSIS FOR DENSITY RATIO ESTIMATE

## E.1    SUPPLEMENTARY MATERIALS ABOUT LOCAL RADEMACHER COMPLEXITY

**Definition E.1** (Rademacher complexity). *Let $\mathcal{F}$ be a class of functions from $\mathcal{X}$ to $\mathbb{R}$ and $\mathcal{X} = \{X_i\}_{i=1}^m \subseteq \mathcal{X}$ be a sample drawn from $\mu_X^m$. Let $\varepsilon = \{\varepsilon_i\}_{i=1}^m$ be independent Rademacher variables. Then the empirical Rademacher complexity of $\mathcal{F}$ with respect to the sample $\mathcal{X}$ is defined as*

$$
\widehat{\mathfrak{R}}_{\mathcal{X}}(\mathcal{F}) = \mathbb{E}_{\varepsilon}\left[\sup_{f\in\mathcal{F}} \frac{1}{m} \sum_{i=1}^m \varepsilon_i f(X_i)\right],
$$

*where $\mathbb{E}_{\varepsilon}[\cdot]$ is the expectation with respect to $\varepsilon$ conditional on $\mathcal{X}$. The Rademacher complexity of $\mathcal{F}$ is the expectation of the empirical Rademacher complexity over all samples drawn according to $\mu_X^m$, that is, $\mathfrak{R}_m = \mathbb{E}_{\mathcal{X}}[\widehat{\mathfrak{R}}_{\mathcal{X}}(\mathcal{F})]$.*

**Lemma E.2** (Lemma A.4 in Bartlett et al. (2005)). *Let $\mathcal{F}$ be a class of functions that map $\mathcal{X}$ into $[-B, B]$ for some positive constant $B$. Then with probability at least $1 - \delta$, the following inequality holds*

$$
\mathfrak{R}_m(\mathcal{F}) \leq 2\widehat{\mathfrak{R}}_{\mathcal{X}}(\mathcal{F}) + \frac{2B\log(1/\delta)}{n}.
$$

**Lemma E.3** (Theorem 2.1 in Bartlett et al. (2005)). *Let $\mathcal{F}$ be a class of functions that map $\mathcal{X}$ into $[-B, B]$ for some positive constant $B$. Assume that there exists some $r > 0$ such that*

$$
\mathcal{F} \subseteq \{f \in \mathcal{F} : \mathbb{E}_X[f^2(X)] \leq r\}.
$$

*Then for each $\delta \in (0, 1)$ with probability at least $1 - \delta$, the following inequality holds*

$$
\sup_{f\in\mathcal{F}} \left(\frac{1}{m}\sum_{i=1}^m f(X_i) - \mathbb{E}_X[f(X)]\right) \leq 3\mathfrak{R}_m(\mathcal{F}) + \sqrt{\frac{2r\log(1/\delta)}{m}} + \frac{14}{3}\frac{B\log(1/\delta)}{m}.
$$

**Lemma E.4** (Corollary 2.2 in Bartlett et al. (2005)). *Let $\mathcal{F}$ be a class of functions that map $\mathcal{X}$ into $[-B, B]$ for some positive constant $B$. For each $\delta \in (0, 1)$ and each $r$ satisfy*

$$
r \geq 12B\mathfrak{R}_m\left(\{f \in \mathcal{F}, \mathbb{E}_X[f^2(X)] \leq r\}\right) + \frac{12B^2\log(1/\delta)}{m},
$$

*the following holds with probability at least $1 - \delta$*

$$
\left\{f \in \mathcal{F} : \mathbb{E}_X[f^2(X)] \leq r\right\} \subseteq \left\{f \in \mathcal{F} : \frac{1}{m}\sum_{i=1}^m f^2(X_i) \leq 2r\right\}.
$$

*Proof.* Note that $\mathbb{E}_X[f^2(X)] \leq r$ implies $\mathbb{E}[f^4(X)] \leq B^2\mathbb{E}[f^2(X)] \leq B^2 r$. Then applying Lemma E.3 gives that for each $f \in \mathcal{F}$ satisfying $\mathbb{E}_X[f^2(X)] \leq r$, the following holds with probability at least $1 - \delta$

$$
\frac{1}{m}\sum_{i=1}^m f^2(X_i) \leq \mathbb{E}_X[f^2(X)] + 3\mathfrak{R}_m\left(\{f^2 : f \in \mathcal{F}, \mathbb{E}_X[f^2(X)] \leq r\}\right)
$$
$$
+ \sqrt{\frac{2B^2r\log(1/\delta)}{m}} + \frac{14}{3}\frac{B^2\log(1/\delta)}{m}
$$
$$
\leq r + 6B\mathfrak{R}_m\left(\{f \in \mathcal{F}, \mathbb{E}_X[f^2(X)] \leq r\}\right) + \frac{r}{2} + \frac{17}{3}\frac{B^2\log(1/\delta)}{m} \leq 2r,
$$

where the second inequality is due to Ledoux-Talagrand contraction inequality (Ledoux & Talagrand, 1991) and Cauchy-Schwarz inequality $ab \leq a^2/4 + b^2$, while the last inequality is owing to the assumption. □

**Definition E.5** (Sub-root function). *A function $\psi : [0, \infty) \to [0, \infty)$ is sub-root if it is non-negative, non-decreasing and if $r \mapsto \psi(r)/\sqrt{r}$ is non-increasing for $r > 0$.*

**Lemma E.6** (Lemma 3.2 in Bartlett et al. (2005)). *If $\psi : [0, \infty) \to [0, \infty)$ is a nontrivial sub-root function, then it is continuous on $[0, \infty)$ and the equality $\psi(r) = r$ has a unique positive solution. Moreover, if we denote the solution by $r^*$, then for each $r > 0$, the inequality $r \geq \psi(r)$ holds if and only if $r^* \leq r$.*

**Lemma E.7** (Bartlett et al. (2005, Theorem 3.3)). *Suppose the following conditions hold:*

(i) *Let $\mathcal{F}$ be a class of functions taking values in $[-B, B]$.*

(ii) *There are some functional $T : \mathcal{F} \to [0, +\infty)$ and some positive constant $V$ such that $\mathbb{E}_X[f^2(X)] \leq T(f) \leq V\mathbb{E}_X[f(X)]$ for each $f \in \mathcal{F}$.*

(iii) *Let $\psi$ be a sub-root function and let $r^*$ be the fixed point of $\psi$, satisfying*

$$\psi(r) \geq V\mathfrak{R}_m\Big(\big\{f \in \mathcal{F} : T(f) \leq r\big\}\Big).$$

*Then for each $\delta \in (0, 1)$, the following holds with probability at least $1 - \delta$*

$$\mathbb{E}_X\big[f(X)\big] \leq \frac{2}{m}\sum_{i=1}^{m} f(X_i) + \frac{1408}{V}r^* + \frac{(22B + 52V)\log(1/\delta)}{m}, \tag{E.1}$$

*for each $f \in \mathcal{F}$. Also, the following holds with probability at least $1 - \delta$*

$$\frac{1}{m}\sum_{i=1}^{m} f(X_i) \leq \frac{3}{2}\mathbb{E}_X\big[f(X)\big] + \frac{1408}{V}r^* + \frac{(22B + 52V)\log(1/\delta)}{m}, \tag{E.2}$$

*for each $f \in \mathcal{F}$.*

To compute the local Rademacher complexities, we introduce Lemma E.8, which is a simplified version of Dudley's integral bound (Srebro & Sridharan, 2010, Theorem 2.1). This lemma is also inspired by Lemma 5.7 in van Handel (2016).

**Lemma E.8** (Lipschitz maximal inequality). *Let $\mathcal{F}$ be a class of functions. The following inequality holds for each $r > 0$*

$$\widehat{\mathfrak{R}}_{\mathcal{X}}\left(\left\{f \in \mathcal{F} : \frac{1}{m}\sum_{i=1}^{m} f^2(X_i) \leq r\right\}\right) \leq \inf_{\varepsilon > 0}\left\{2\varepsilon + \sqrt{\frac{2r\log N(\varepsilon, \mathcal{F}, L^\infty(\mathcal{X}))}{m}}\right\}.$$

Before the proof of Lemma E.8, we first introduce Massart's lemma as preparation.

**Lemma E.9** (Massart's lemma). *Let $\mathcal{F}$ be a class of functions satisfying $|\mathcal{F}| < \infty$. The following inequality holds for each $r > 0$*

$$\widehat{\mathfrak{R}}_{\mathcal{X}}\left(\left\{f \in \mathcal{F} : \frac{1}{m}\sum_{i=1}^{m} f^2(X_i) \leq r\right\}\right) \leq \sqrt{\frac{2r\log|\mathcal{F}|}{m}}.$$

*Proof.* Let $\{\varepsilon_i\}_{i=1}^m$ be a set of independent Rademacher random variables. For the fixed sample $\mathcal{X} = \{X_i\}_{i=1}^n$, $\{\varepsilon_i f(X_i)\}_{i=1}^m$ are random variables satisfying $-f(X_i) \leq \varepsilon_i f(X_i) \leq f(X_i)$ and $\mathbb{E}[\varepsilon_i f(X_i)|X_i] = 0$ for $i \in [n]$. Then it follows from Hoeffding's lemma (Mohri et al., 2018, Lemma D.1) that for each $i \in [n]$,

$$\mathbb{E}\big[\exp(\lambda\varepsilon_i f(X_i))\big|X_i\big] \leq \exp\Big(\frac{\lambda^2 f^2(X_i)}{2}\Big).$$

Consequently, we have

$$
\mathbb{E}\Big[\exp\Big(\lambda\sum_{i=1}^{m}\varepsilon_i f(X_i)\Big)\Big|\mathcal{X}\Big] = \mathbb{E}\Big[\prod_{i=1}^{m}\exp(\lambda\varepsilon_i f(X_i))\Big|\mathcal{X}\Big] = \prod_{i=1}^{m}\mathbb{E}\big[\exp(\lambda\varepsilon_i f(X_i))\big|X_i\big]
$$

$$
\leq \prod_{i=1}^{m}\exp\Big(\frac{\lambda^2 f^2(X_i)}{2}\Big) = \exp\Big(\frac{\lambda^2 \sum_{i=1}^{m} f^2(X_i)}{2}\Big).
$$

Furthermore, it follows from Jensen's inequality that

$$
\exp\Big(\lambda\mathbb{E}\Big[\max_{f\in\mathcal{F}}\sum_{i=1}^{m}\varepsilon_i f(X_i)\Big|\mathcal{X}\Big]\Big)
$$

$$
\leq \mathbb{E}\Big[\exp\Big(\lambda\max_{f\in\mathcal{F}}\sum_{i=1}^{m}\varepsilon_i f(X_i)\Big)\Big|\mathcal{X}\Big] = \mathbb{E}\Big[\max_{f\in\mathcal{F}}\exp\Big(\lambda\sum_{i=1}^{m}\varepsilon_i f(X_i)\Big)\Big|\mathcal{X}\Big]
$$

$$
\leq \mathbb{E}\Big[\sum_{f\in\mathcal{F}}\exp\Big(\lambda\sum_{i=1}^{m}\varepsilon_i f(X_i)\Big)\Big|\mathcal{X}\Big] \leq |\mathcal{F}|\exp\Big(\frac{\lambda^2 \sum_{i=1}^{m} f^2(X_i)}{2}\Big).
$$

Taking the logarithm of both sides of the inequality yields

$$
\mathbb{E}\Big[\max_{f\in\mathcal{F}}\sum_{i=1}^{m}\varepsilon_i f(X_i)\Big|\mathcal{X}\Big] \leq \frac{\log|\mathcal{F}|}{\lambda} + \frac{\lambda\sum_{i=1}^{m} f^2(X_i)}{2}.
$$

Setting $\lambda^2 = 2\log|\mathcal{F}|(\sum_{i=1}^{m} f^2(X_i))^{-1}$ gives

$$
\widehat{\mathfrak{R}}_{\mathcal{X}}(\mathcal{F}) \leq \mathbb{E}\Big[\max_{f\in\mathcal{F}}\sum_{i=1}^{m}\varepsilon_i f(X_i)\Big|\mathcal{X}\Big] \leq \sqrt{\frac{2(\frac{1}{m}\sum_{i=1}^{m} f^2(X_i))\log|\mathcal{F}|}{m}},
$$

which completes the proof. $\qquad\square$

*Proof of Lemma E.8.* Denote $\mathcal{F}^r = \{f\in\mathcal{F}: \frac{1}{m}\sum_{i=1}^{m} f^2(X_i)\leq r\}$. Let $\mathcal{F}_\varepsilon^r$ be an $L^\infty(\mathcal{X})$ $\varepsilon$-cover of $\mathcal{F}^r$ such that $|\mathcal{F}_\varepsilon^r| = N(\varepsilon,\mathcal{F}^r,L^\infty(\mathcal{X}))$, which means, for each $f\in\mathcal{F}^r$ there exists $f_\varepsilon\in\mathcal{F}_\varepsilon^r$ such that $\max_{1\leq i\leq n}|f(X_i) - f_\varepsilon(X_i)|\leq\varepsilon$. For $f_\varepsilon\in\mathcal{F}_\varepsilon$, if $\frac{1}{m}\sum_{i=1}^{m} f_\varepsilon^2(X_i)\leq r$, we define $\tilde{f}_\varepsilon = f_\varepsilon$. If $\frac{1}{m}\sum_{i=1}^{m} f_\varepsilon^2(X_i) > r$, let $\tilde{f}_\varepsilon$ be the nearest element of $f_\varepsilon$ in $\mathcal{F}^r$, that is,

$$
\tilde{f}_\varepsilon \in \operatorname*{arg\,min}_{f\in\mathcal{F}^r}\Big(\max_{1\leq i\leq n}|f(X_i) - f_\varepsilon(X_i)|\Big).
$$

Then it is apparent that for each $f\in\mathcal{F}^r$,

$$
\max_{1\leq i\leq n}|f_\varepsilon(X_i) - \tilde{f}_\varepsilon(X_i)| \leq \max_{1\leq i\leq n}|f_\varepsilon(X_i) - f(X_i)| \leq \varepsilon.
$$

According to the triangular inequality, for each $f\in\mathcal{F}^r$ satisfying $\max_{1\leq i\leq n}|f(X_i) - f_\varepsilon(X_i)|\leq\varepsilon$, it holds that

$$
\max_{1\leq i\leq n}|f(X_i) - \tilde{f}_\varepsilon(X_i)| \leq \max_{1\leq i\leq n}|f(X_i) - f_\varepsilon(X_i)| + \max_{1\leq i\leq n}|f_\varepsilon(X_i) - \tilde{f}_\varepsilon(X_i)| \leq 2\varepsilon.
$$

Hence $\tilde{\mathcal{F}}_\varepsilon^r = \{\tilde{f}_\varepsilon : f_\varepsilon\in\mathcal{F}_\varepsilon^r\}$ is an $L^\infty(\mathcal{X})$ $(2\varepsilon)$-cover of $\mathcal{F}^r$ satisfying $|\tilde{\mathcal{F}}_\varepsilon^r| = N(\varepsilon,\mathcal{F}^r,L^\infty(\mathcal{X}))$, and $\frac{1}{m}\sum_{i=1}^{m}\tilde{f}_\varepsilon^2(X_i)\leq r$ for each $\tilde{f}_\varepsilon\in\tilde{\mathcal{F}}_\varepsilon^r$. Then it is straightforward that

$$
\widehat{\mathfrak{R}}_{\mathcal{X}}(\mathcal{F}^r) = \mathbb{E}\Big[\sup_{f\in\mathcal{F}^r}\frac{1}{m}\sum_{i=1}^{m}\varepsilon_i(f(X_i) - \tilde{f}_\varepsilon(X_i))\Big|\mathcal{X}\Big] + \mathbb{E}\Big[\sup_{\tilde{f}_\varepsilon\in\tilde{\mathcal{F}}_\varepsilon^r}\frac{1}{m}\sum_{i=1}^{m}\varepsilon_i\tilde{f}_\varepsilon(X_i)\Big|\mathcal{X}\Big]
$$

$$
\leq \sup_{f\in\mathcal{F}^r}\max_{1\leq i\leq n}|f(X_i) - \tilde{f}_\varepsilon(X_i)| + \widehat{\mathfrak{R}}_{\mathcal{X}}(\tilde{\mathcal{F}}_\varepsilon^r) \leq 2\varepsilon + \widehat{\mathfrak{R}}_{\mathcal{X}}(\tilde{\mathcal{F}}_\varepsilon^r),
$$

where the first inequality holds from Hölder's inequality. Combining this with Lemma E.9 and noting that $N(\varepsilon,\mathcal{F}^r,L^\infty(\mathcal{X})) \leq N(\varepsilon,\mathcal{F},L^\infty(\mathcal{X}))$ yield the desired result. $\qquad\square$

**Lemma E.10.** *Let $\mathcal{F}$ be a class of functions, and $f^*$ be a function may depending on $\mathcal{X}$. Then it follows that*

$$\widehat{\mathfrak{R}}_{\mathcal{X}}(\mathcal{F}) = \widehat{\mathfrak{R}}_{\mathcal{X}}(\mathcal{F} - f^*),$$

*where $\mathcal{F} - f^* = \{f - f^* : f \in \mathcal{F}\}$.*

*Proof of Lemma E.10.* It is straightforward that

$$\widehat{\mathfrak{R}}_{\mathcal{X}}(\mathcal{F} - f^*) = \mathbb{E}_{\varepsilon}\left[\sup_{f \in \mathcal{F}} \frac{1}{m} \sum_{i=1}^{m} \varepsilon_i (f(X_i) - f^*(X_i))\right]$$

$$= \mathbb{E}_{\varepsilon}\left[\sup_{f \in \mathcal{F}} \frac{1}{m} \sum_{i=1}^{m} \varepsilon_i f(X_i)\right] - \mathbb{E}_{\varepsilon}\left[\frac{1}{m} \sum_{i=1}^{m} \varepsilon_i f^*(X_i)\right] = \widehat{\mathfrak{R}}_{\mathcal{X}}(\mathcal{F}),$$

which completes the proof. $\qquad\square$

**Definition E.11** (Star-hull). Let $\mathcal{F}$ be a class of functions mapping $\mathcal{X}$ to $\mathbb{R}$. The star-hull of $\mathcal{F}$ around $f^* : \mathcal{X} \to \mathbb{R}$ is defined by

$$\mathrm{star}(\mathcal{F}, f^*) = \big\{ f^* + \alpha(f - f^*) : f \in \mathcal{F}, \alpha \in [0,1] \big\}.$$

Notice that making a class star-hull increases the complexities. However, this increase is moderate as shown in the following lemma.

**Lemma E.12** (Lemma 4.5 in Mendelson (2002)). *Let $f^* : \mathcal{X} \to [-B, B]$ and $\mathcal{F}$ be a class of functions that map $\mathcal{X}$ into $[-B, B]$ for some positive constant $B$. Then the following inequality holds for each $\varepsilon > 0$*

$$\log N(\varepsilon, \mathrm{star}(\mathcal{F}, f^*), L^\infty(\mathcal{X})) \leq \log N(\varepsilon/2, \mathcal{F}, L^\infty(\mathcal{X})) + \log(4B/\varepsilon).$$

*Proof.* Let $\mathcal{F}_\varepsilon$ be an $L^\infty(\mathcal{X})$ $(\varepsilon/2)$-cover of $\mathcal{F}$ such that

$$N = |\mathcal{F}_\varepsilon| = N(\varepsilon/2, \mathcal{F}, L^\infty(\mathcal{X})).$$

Denote $\mathcal{F}_\varepsilon = \{f_j\}_{j=1}^N$. Without loss of generality, we assume that $|f_j(X_i)| \leq B$ for each $i \in [m]$. Then for each $f \in \mathcal{F}$ there exists $j \in [N]$ such that $\max_{1 \leq i \leq n} |f(X_i) - f_j(X_i)| \leq \varepsilon/2$. Denote by $I(f^*, f_j)$ the segment between $f^*$ and $f_j$:

$$I(f^*, f_j) = \big\{ f^* + \alpha(f_j - f^*) : \alpha \in [0,1] \big\}.$$

Furthermore, we construct an $L^\infty(\mathcal{X})$ $(\varepsilon/(4B))$-cover of it by

$$I_\varepsilon(f^*, f_j) = \left\{ f^* + \alpha_k(f_j - f^*) : \alpha_k = \frac{k\varepsilon}{4B}, k = 1, \ldots, \left\lfloor \frac{4B}{\varepsilon} \right\rfloor \right\}.$$

Observe that $\cup_{j=1}^N I(f^*, f_j)$ is an $L^\infty(\mathcal{X})$ $\varepsilon$-cover of $\mathrm{star}(\mathcal{F}, f^*)$. Indeed, it holds that

$$\max_{1 \leq i \leq n} |f^*(X_i) + \alpha(f(X_i) - f^*(X_i)) - f^*(X_i) - \alpha_k(f_j(X_i) - f^*(X_i))|$$

$$\leq |\alpha - \alpha_k| \max_{1 \leq i \leq n} |f^*(X_i)| + \alpha \max_{1 \leq i \leq n} |f(X_i) - f_j(X_i)| + |\alpha - \alpha_k| \max_{1 \leq i \leq n} |f_j(X_i)|$$

$$\leq \frac{\varepsilon}{4B} B + \frac{\varepsilon}{2} + \frac{\varepsilon}{4B} B = \varepsilon.$$

Therefore, it follows that $N(\varepsilon, \mathrm{star}(\mathcal{F}, f^*), L^\infty(\mathcal{X})) \leq N(\varepsilon/2, \mathcal{F}, L^\infty(\mathcal{X}))4B/\varepsilon$, which completes the proof. $\qquad\square$

### E.2 ORACLE INEQUALITY OF DENSITY RATIO ESTIMATOR

To begin with, we have $\mathcal{S} = \{(X_i^\mu, Z_i^\mu)\}_{i=1}^m$ i.i.d. drawn from the probability distribution $\mu$, which denotes the distribution of $(X^\mu, Z^\mu)$ defined in Remark 2.2. Let $\mu_X$ be the marginal distribution of $X^\mu$. It is easy to verify that $2\|u\|_{L^2(\mu_X)}^2 = \|u\|_{L^2(P_X)}^2 + \|u\|_{L^2(Q_X)}^2$. Let $u^* = -\log \varrho$ and define the pre-training excess risk $R^{\mathrm{pre}}(u)$ by $R^{\mathrm{pre}}(u) = L^{\mathrm{pre}}(u) - L^{\mathrm{pre}}(u^*)$.

**Lemma E.13.** *Suppose Assumptions 2, 5 and 7 hold. Let $u^* = -\log \varrho$. Then it follows that*

$$\frac{1}{2}\min\left\{\frac{\Lambda}{(1+\Lambda)^2}, \frac{\lambda}{(1+\lambda)^2}\right\}\|u - u^*\|_{L^2(\mu_X)}^2 \le L^{\mathrm{pre}}(u) - L^{\mathrm{pre}}(u^*) \le \frac{1}{8}\|u - u^*\|_{L^2(\mu_X)}^2.$$

*Proof.* Denote $\ell_{\mathrm{logit}}(v, z) = \log(1 + \exp(-zv))$. According to Taylor's expansion, we find that

$$\ell_{\mathrm{logit}}(u(X), Z) - \ell_{\mathrm{logit}}(u^*(X), Z)$$
$$= -\frac{u(X) - u^*(X)}{1 + \exp(Zu^*(X))} + \frac{1}{2}\frac{\exp(Z\theta(X))}{(1 + \exp(Z\theta(X)))^2}(u(X) - u^*(X))^2,$$

where $\theta(x) = cu(x) + (1 - c)u^*(x)$ for some $c \in [0, 1]$. Taking expectation with respect to $(X, Z) \sim \mu$ yields

$$L^{\mathrm{pre}}(u) - L^{\mathrm{pre}}(u^*) = \mathbb{E}_{(X,Z)\sim\mu}\left[\frac{1}{2}\frac{\exp(Z\theta(X))}{(1 + \exp(Z\theta(X)))^2}(u(X) - u^*(X))^2\right],$$

where we used the fact that $u^*$ is the minimizer of $L^{\mathrm{pre}}(\cdot)$ over $\mathscr{L}(\mathcal{X})$. According to Assumptions 2 and 7, the following inequality holds for each $x \in \mathcal{X}$

$$\min\left\{\frac{\Lambda}{(1+\Lambda)^2}, \frac{\lambda}{(1+\lambda)^2}\right\} \le \frac{\exp(\theta(x))}{(1 + \exp(\theta(x)))^2} \le \frac{1}{4},$$

which completes the proof. □

**Lemma E.14.** *Suppose Assumptions 2, 5 and 7 hold. Let $u^* = -\log \varrho$. Let $\mathcal{S} = \{(X_i^\mu, Z_i^\mu)\}_{i=1}^m$ be an i.i.d. sample set drawn from $\mu$. Let $\hat{u}_{\mathcal{S}} \in \mathcal{U}$ be defined in (2.9). Define the distribution-dependent measurement functional*

$$T(u) = \mathbb{E}_{X^\mu\sim\mu_X}\left[(u(X^\mu) - u^*(X^\mu))^2\right].$$

*Assume that $\psi$ ia a sub-root function for which*

$$\psi(r) \ge V\Re_m\left(\{u \in \mathcal{U} : T(u) \le r\}\right).$$

*Further, let $r^*$ be the fixed point of $\psi$. Then for each $\delta \in (0, 1)$ the following inequality holds with probability as least $1 - 2\delta$*

$$R^{\mathrm{pre}}(\hat{u}_{\mathcal{S}}) \le 3\inf_{u\in\mathcal{U}} R^{\mathrm{pre}}(u) + \frac{4224}{V}r^* + \frac{(132M + 156V)\log(1/\delta)}{m},$$

*where $V = \frac{1}{2}\max\{\frac{(1+\Lambda)^2}{\Lambda}, \frac{(1+\lambda)^2}{\lambda}\}$ and $M = \max\{\log(1 + 1/\lambda), \log(1 + \Lambda)\}$.*

*Proof of Lemma E.14.* For simplicity of notations, we define

$$g(u, x, z) = \ell_{\mathrm{logit}}(u(x), z) - \ell_{\mathrm{logit}}(u^*(x), z), \quad \ell_{\mathrm{logit}}(v, z) = \log(1 + \exp(-zv)).$$

Notice that the pre-training excess risk satisfies $R^{\mathrm{pre}}(u) = \mathbb{E}_{(X,Z)\sim\mu}[g(u, X, Z)]$. Since $\ell_{\mathrm{logit}}(u(x), z)$ is 1-Lipschitz with respect to $u(x)$ for each $z \in \{\pm 1\}$, which deduces that the following inequality holds for each $(x, z) \in \mathcal{X} \times \{\pm 1\}$

$$|g(u, x, z)| = |\ell_{\mathrm{logit}}(u(x), z) - \ell_{\mathrm{logit}}(u^*(x), z)| \le |u(x) - u^*(x)|.$$

As a consequence, it holds that

$$\mathbb{E}_{(X^\mu,Z^\mu)\sim\mu}\left[g^2(u, X^\mu, Z^\mu)\right] \le \|u - u^*\|_{L^2(\mu_X)}^2 = T(u), \tag{E.3}$$

where $T(u)$ is called the measurement functional. On the other hand, using Lemma E.13 yields

$$T(u) \le V R^{\mathrm{pre}}(u) = V\mathbb{E}_{(X^\mu,Z^\mu)\sim\mu}\left[g(u, X^\mu, Z^\mu)\right]. \tag{E.4}$$

By Ledoux-Talagrand contraction inequality (Ledoux & Talagrand, 1991), we have

$$\Re_m\left(\{g \circ u : u \in \mathcal{U}, T(u) \le r\}\right) \le \Re_m\left(\{u \in \mathcal{U} : T(u) \le r\}\right), \tag{E.5}$$

which implies that

$$\psi(r) \geq V \mathfrak{R}_m \Big( \{ g \circ u : u \in \mathcal{U}, T(u) < r \} \Big).$$

By applying (E.1) in Lemma E.7 to the function $g \circ u$, (E.3) to (E.5) deduces that the following inequality holds with probability as least $1 - \delta$ for each $u \in \mathcal{U}$

$$R^{\text{pre}}(u) \leq 2 \widehat{R}_{\mathcal{S}}^{\text{pre}}(u) + \frac{1408}{V} r^* + \frac{(44M + 52V) \log(1/\delta)}{m},$$

where we used the fact that $|g(u, x, z)| \leq |u(x) - u^*(x)| \leq 2M$ for each $(x, z) \in \mathcal{X} \times \{\pm 1\}$ from Assumptions 2 and 7. Since $\hat{u}_{\mathcal{S}}$ is the minimizer of $\widehat{R}_{\mathcal{S}}^{\text{pre}}(\cdot)$ over $\mathcal{U}$, we have that with probability as least $1 - \delta$ for each $u \in \mathcal{U}$,

$$R^{\text{pre}}(\hat{u}_{\mathcal{S}}) \leq 2 \widehat{R}_{\mathcal{S}}^{\text{pre}}(u) + \frac{1408}{V} r^* + \frac{(44M + 52V) \log(1/\delta)}{m}, \tag{E.6}$$

Further, using (E.2) in Lemma E.7 gives that the following inequality holds with with probability as least $1 - \delta$ for each $u \in \mathcal{U}$,

$$\widehat{R}_{\mathcal{S}}^{\text{pre}}(u) \leq \frac{3}{2} R^{\text{pre}}(u) + \frac{1408}{V} r^* + \frac{(44M + 52V) \log(1/\delta)}{m}. \tag{E.7}$$

Combining (E.6) and (E.7) yields the desired result. □

The results in Lemma E.14 use distribution-dependent measures of complexity of the class. By a similar technique as the proof of Bartlett et al. (2005, Lemma 3.4), we next provide error bounds which can be identified directly from the sample set, without a priori information.

**Lemma E.15.** *Suppose Assumptions 2, 5 and 7 hold. Let $u^* = -\log \varrho$. Let $\mathcal{S} = \{(X_i^\mu, Z_i^\mu)\}_{i=1}^m$ be an i.i.d. sample set drawn from $\mu$. Let $\hat{u}_{\mathcal{S}} \in \mathcal{U}$ be defined in (2.9). Define the data-dependent measurement functional*

$$\widehat{T}_{\mathcal{S}}(u) = \frac{1}{m} \sum_{i=1}^m (u(X_i^\mu) - u^*(X_i^\mu))^2$$

*Assume that $\widehat{\psi}_{\mathcal{S}}$ ia a sub-root function for which*

$$\widehat{\psi}_{\mathcal{S}}(r) \geq 2(12M + V) \widehat{\mathfrak{R}}_{\mathcal{S}} \Big( \{ u \in \text{star}(\mathcal{U}, u^*) : \widehat{T}_{\mathcal{S}}(u) \leq 2r \} \Big) + \frac{(36M^2 + 2VM) \log(1/\delta)}{m}.$$

*Further, let $\hat{r}_{\mathcal{S}}^*$ be the fixed point of $\widehat{\psi}_{\mathcal{S}}$ and $r \geq \hat{r}_{\mathcal{S}}^*$. Then for each $\delta \in (0, 1)$ the following inequality holds with probability at least $1 - 4\delta$*

$$R^{\text{pre}}(\hat{u}_{\mathcal{S}}) \leq 3 \inf_{u \in \mathcal{U}} R^{\text{pre}}(u) + \frac{4224}{V} r + \frac{(132M + 156V) \log(1/\delta)}{m},$$

*where $V = \frac{1}{2} \max\{ \frac{(1+\Lambda)^2}{\Lambda}, \frac{(1+\lambda)^2}{\lambda} \}$ and $M = \max\{\log(1 + 1/\lambda), \log(1 + \Lambda)\}$.*

*Proof.* We use the same definition of the measurement functional $T(u)$ as in Lemma E.14. The proof is divided into two steps.

*Step 1. Construct a non-trivial sub-root function.*

Let $\psi$ be a sub-root function satisfying

$$\psi(r) \geq V \mathfrak{R}_m \Big( \{ u \in \mathcal{U} : T(u) \leq r \} \Big), \tag{E.8}$$

and

$$\psi(r) \geq 12M \mathfrak{R}_m \Big( \{ u \in \mathcal{U} : T(u) \leq r \} \Big) + \frac{12M^2 \log(1/\delta)}{m}. \tag{E.9}$$

To this end, we set $\psi$ as

$$\psi(r) = (12M + V) \mathfrak{R}_m \Big( \{ u \in \text{star}(\mathcal{U}, u^*) : T(u) \leq r \} \Big) + \frac{12M^2 \log(1/\delta)}{m}. \tag{E.10}$$

Now we show that $\psi$ is a sub-root function. By Jensen's inequality, we find that the Rademacher complexity is non-negative and thus $\psi$ is non-negative. Furthermore, $\psi$ is non-decreasing since the following holds for $r \le r'$

$$\{u \in \text{star}(\mathcal{U}, u^*) : T(u) \le r\} \subseteq \{u \in \text{star}(\mathcal{U}, u^*) : T(u) \le r'\}.$$

It remains to show that for each $0 < r_1 < r_2$, it holds that $\psi(r_1) \ge \sqrt{r_1/r_2}\psi(r_2)$. According to (E.10) and Lemma E.10, we only need to verify

$$
\Re_m\Big(\{v \in \text{star}(\mathcal{U} - u^*, 0) : \mathbb{E}_{X^\mu \sim \mu_X}[v^2(X^\mu)] \le r_1\}\Big)
$$
$$
\ge \sqrt{\frac{r_1}{r_2}}\Re_m\Big(\{v \in \text{star}(\mathcal{U} - u^*, 0) : \mathbb{E}_{X^\mu \sim \mu_X}[v^2(X^\mu)] \le r_2\}\Big). \tag{E.11}
$$

Fix a sample set $\{X_i\}_{i=1}^m$ drawn from $\mu_X$ and a set of Rademacher variables $\{\varepsilon_i\}_{i=1}^m$, define

$$\eta = \sup\Big\{\frac{1}{m}\sum_{i=1}^m \varepsilon_i v(X_i^\mu) : v \in \text{star}(\mathcal{U} - u^*, 0), \mathbb{E}_{X \sim \mu_X}[v^2(X^\mu)] \le r_2\Big\}.$$

Let $\{v_k\}_{k=1}^\infty \subseteq \{v \in \text{star}(\mathcal{U} - u^*, 0), \mathbb{E}_{X \sim \mu_X}[v^2(X^\mu)] \le r_2\}$ be a sequence of functions for which

$$\eta = \lim_{k \to \infty} \frac{1}{n}\sum_{i=1}^m \varepsilon_i v_k(X_i^\mu) \quad \text{and} \quad \eta \ge \frac{1}{m}\sum_{i=1}^m \varepsilon_i v_k(X_i^\mu), \quad k \ge 1$$

Since $\mathbb{E}_{X^\mu \sim \mu_X}[v_k^2(X^\mu)] \le r_2$, we find that $\mathbb{E}_{X^\mu \sim \mu_X}[(\sqrt{r_1/r_2}v_k(X^\mu))^2] \le r_1$. Moreover, by the definition of star-hull, we find that $\sqrt{r_1/r_2}v_k \in \text{star}(\mathcal{U} - u^*, 0)$ for each $k \ge 1$. Thus the following inequality holds for each $k \ge 1$

$$\sup\Big\{\frac{1}{m}\sum_{i=1}^m \varepsilon_i v(X_i^\mu) : v \in \text{star}(\mathcal{U} - u^*, 0), \mathbb{E}_{X^\mu \sim \mu_X}[v^2(X^\mu)] \le r_1\Big\} \ge \sqrt{\frac{r_1}{r_2}}\frac{1}{m}\sum_{i=1}^m \varepsilon_i v_k(X_i^\mu).$$

Then taking limitation as $k \to \infty$, we find that

$$\sup\Big\{\frac{1}{m}\sum_{i=1}^m \varepsilon_i v(X_i^\mu) : v \in \text{star}(\mathcal{U} - u^*, 0), \mathbb{E}_{X^\mu \sim \mu_X}[v^2(X^\mu)] \le r_1\Big\} \ge \sqrt{\frac{r_1}{r_2}}\eta.$$

Taking expectation with respect to the sample set $\{X_i^\mu\}_{i=1}^m$ and Rademacher variables $\{\varepsilon_i\}_{i=1}^n$ yields (E.11). Hence we conclude that $\psi(r)$ is a non-trivial sub-root function.

*Step 2. Data-dependent error bounds.*

Let $r^*$ be the fixed point of the sub-root function $\psi$ defined as (E.10). Since $\psi$ satisfies (E.8), using Lemma E.14 implies that for $\delta \in (0, 1)$ the following inequality holds with probability as least $1 - 2\delta$

$$R^{\text{pre}}(\hat{u}_\mathbb{S}) \le 3\inf_{u \in \mathcal{U}} R^{\text{pre}}(u) + \frac{4224}{V}r^* + \frac{(132M + 156V)\log(1/\delta)}{m}. \tag{E.12}$$

Notice that (E.12) uses distribution-dependent measures of complexity of the function class. We next establish distribution-free bounds, which only depend on a sample set $\mathbb{S}$. Since that $\psi$ satisfies (E.9), applying Lemma E.4 gives that with probability at least $1 - \delta$,

$$\{u \in \text{star}(\mathcal{U}, u^*) : T(u) \le r^*\} \subseteq \{u \in \text{star}(\mathcal{U}, u^*) : \widehat{T}_\mathbb{S}(u) \le 2r^*\},$$

which means that the following inequality holds with probability at least $1 - \delta$

$$\widehat{\Re}_\mathbb{S}\Big(\{u \in \text{star}(\mathcal{U}, u^*) : T(u) \le r^*\}\Big) \le \widehat{\Re}_\mathbb{S}\Big(\{u \in \text{star}(\mathcal{U}, u^*) : \widehat{T}_\mathbb{S}(u) \le 2r^*\}\Big). \tag{E.13}$$

In addition, we find from Lemma E.2 that the following inequality holds with probability at least $1 - \delta$

$$
\Re_m\Big(\{u \in \text{star}(\mathcal{U}, u^*) : T(u) \le r^*\}\Big)
$$
$$
\le 2\widehat{\Re}_\mathbb{S}\Big(\{u \in \text{star}(\mathcal{U}, u^*) : T(u) \le r^*\}\Big) + \frac{2M\log(1/\delta)}{m}. \tag{E.14}
$$

Combining (E.10), (E.13) and (E.14) obtains that the following inequality holds with probability at least $1 - 2\delta$

$$
\begin{aligned}
\psi(r^*) &\leq 2(12M + V)\widehat{\mathfrak{R}}_{\mathcal{S}}\Big(\big\{u \in \mathrm{star}(\mathcal{U}, u^*) : T(u) \leq r^*\big\}\Big) + \frac{(36M^2 + 2VM)\log(1/\delta)}{m} \\
&\leq 2(12M + V)\widehat{\mathfrak{R}}_{\mathcal{S}}\Big(\big\{u \in \mathrm{star}(\mathcal{U}, u^*) : \widehat{T}_{\mathcal{S}}(u) \leq 2r^*\big\}\Big) + \frac{(36M^2 + 2VM)\log(1/\delta)}{m} \\
&\leq \widehat{\psi}_{\mathcal{S}}(r^*).
\end{aligned}
$$

As a consequence, we have $r^* = \psi(r^*) \leq \widehat{\psi}_{\mathcal{S}}(r^*)$, which deduces $r^* \leq \hat{r}_{\mathcal{S}}^*$ with probability $1 - 2\delta$ from Lemma E.6. Recalling that (E.12) holds with probability at least $1 - 2\delta$ yields the desired result. $\qquad\square$

**Lemma E.16.** *Suppose Assumptions 2, 5 and 7 hold. Let $u^* = -\log \varrho$. Let $\mathcal{S} = \{(X_i^\mu, Z_i^\mu)\}_{i=1}^m$ be an i.i.d. sample set drawn from $\mu$. Let $\hat{u}_{\mathcal{S}} \in \mathcal{U}$ be defined as (2.9). Then the following inequality holds*

$$
\mathbb{E}_{\mathcal{S}}\big[R^{\mathrm{pre}}(\hat{u}_{\mathcal{S}})\big] \lesssim \inf_{u \in \mathcal{U}} R^{\mathrm{pre}}(u) + \frac{M^2 + V^2}{V} \frac{\mathrm{VCdim}(\mathcal{U})\log(em)}{m},
$$

*where $V = \frac{1}{2}\max\{\frac{(1+\Lambda)^2}{\Lambda}, \frac{(1+\lambda)^2}{\lambda}\}$ and $M = \max\{\log(1+1/\lambda), \log(1+\Lambda)\}$.*

*Proof of Lemma E.16.* We divide the proof into two steps.

*Step 1. Oracle inequality with high-probability statement.*

Using Lemmas E.8 and E.12 and setting $\varepsilon = M/m$, we find that for $n \geq \mathrm{VCdim}(\mathcal{F})$,

$$
\begin{aligned}
\widehat{\mathfrak{R}}_{\mathcal{S}}\Big(\big\{u \in \mathrm{star}(\mathcal{U}, u^*) : \widehat{T}_{\mathcal{S}}(u) \leq 2r\big\}\Big) &\leq \frac{2M}{m} + 2\sqrt{\frac{\log\{4mN(M/(2m), \mathcal{U}, L^\infty(\mathcal{S}))\}}{m}}\sqrt{r} \\
&\leq \frac{2M}{m} + 4\sqrt{\frac{\mathrm{VCdim}(\mathcal{U})\log(em)}{m}}\sqrt{r},
\end{aligned}
$$

where the second inequality is due to Anthony et al. (1999, Theorem 12.2). Define the sub-root function $\widehat{\psi}_{\mathcal{S}}(r)$ by $\widehat{\psi}_{\mathcal{S}}(r) = a\sqrt{r} + b$, where

$$
a = 8(12M + V)\sqrt{\frac{\mathrm{VCdim}(\mathcal{U})\log(em)}{m}} \quad \text{and} \quad b = \frac{(60M^2 + 4VM)\log(1/\delta)}{m}.
$$

By setting $r = 4a^2 + 2b$, we find that $\widehat{\psi}_{\mathcal{S}}(r) \leq r$. Combining this with Lemma E.6 implies $\hat{r}_{\mathcal{S}}^* \leq r$. Then using Lemma E.15 yields

$$
\Pr(R^{\mathrm{pre}}(\hat{u}_{\mathcal{S}}) > \varepsilon(\delta, n)) \leq 4\delta,
$$

where

$$
\varepsilon(\delta, n) = 3\inf_{u \in \mathcal{U}} R^{\mathrm{pre}}(u) + \frac{4224(4a^2 + 2b)}{V} + \frac{(132M + 156V)\log(1/\delta)}{m}.
$$

*Step 2. Convergence rates of the density ratio estimator.*

Since $g(u, x, z) \leq 2M$ for each $u \in \mathcal{U}$ and $(x, z) \in \mathcal{X} \times \{\pm 1\}$, it follows that $R^{\mathrm{pre}}(\hat{u}_{\mathcal{S}}) \leq 2M$, and consequently,

$$
\begin{aligned}
&\mathbb{E}_{\mathcal{S}}\Big[R^{\mathrm{pre}}(\hat{u}_{\mathcal{S}})\Big] \\
&= \mathbb{E}_{\mathcal{S}}\Big[R^{\mathrm{pre}}(\hat{u}_{\mathcal{S}}) \cdot \mathbb{I}(R^{\mathrm{pre}}(\hat{u}_{\mathcal{S}}) > \varepsilon(\delta, n))\Big] + \mathbb{E}_{\mathcal{S}}\Big[R^{\mathrm{pre}}(\hat{u}_{\mathcal{S}}) \cdot \mathbb{I}(R^{\mathrm{pre}}(\hat{u}_{\mathcal{S}}) \leq \varepsilon(\delta, n))\Big] \\
&\leq 2M \cdot \Pr(R^{\mathrm{pre}}(\hat{u}_{\mathcal{S}}) > \varepsilon(\delta, n)) + \varepsilon(\delta, n) \cdot (1 - \Pr(R^{\mathrm{pre}}(\hat{u}_{\mathcal{S}}) > \varepsilon(\delta, n))) \\
&\leq 8M\delta + \varepsilon(\delta, n),
\end{aligned}
$$

Setting $\delta = 1/m$ completes the proof. $\qquad\square$

**Lemma E.17** (Oracle inequality of density-ratio estimator). *Suppose Assumptions 2, 5 and 7 hold. Let $\mathcal{S} = \{(X_i^\mu, Z_i^\mu)\}_{i=1}^m$ be an i.i.d. sample set drawn from (2.8). Suppose that $\mathcal{U}$ is a hypothesis class and $\hat{u}_\mathcal{S} \in \mathcal{U}$ is defined by (2.9). Then the following inequality holds for $m \geq \mathrm{VCdim}(\mathcal{U})$,*

$$\mathbb{E}_{\mathcal{S} \sim \mu^m}\left[\|\hat{\varrho}_\mathcal{S} - \varrho\|_{L^2(P_X)}^2 + \|\hat{\varrho}_\mathcal{S} - \varrho\|_{L^2(Q_X)}^2\right]$$

$$\lesssim \inf_{u \in \mathcal{U}}\left(\|u + \log \varrho\|_{L^2(P_X)}^2 + \|u + \log \varrho\|_{L^2(Q_X)}^2\right) + \frac{M^2 + V^2}{V}\,\mathrm{VCdim}(\mathcal{U})\frac{\log(em)}{m},$$

*where $V = \frac{1}{2}\max\{\frac{(1+\Lambda)^2}{\Lambda}, \frac{(1+\lambda)^2}{\lambda}\}$ and $M = \max\{\log(1 + 1/\lambda), \log(1 + \Lambda)\}$.*

*Proof of Lemma E.17.* Observe that

$$\begin{aligned}
\|\hat{\varrho}_\mathcal{S} - \varrho\|_{L^2(\mu_X)}^2 &= \mathbb{E}_{X^\mu \sim \mu_X}\left[(\hat{\varrho}_\mathcal{S}(X^\mu) - \varrho(X^\mu))^2\right] \\
&= \mathbb{E}_{X^\mu \sim \mu_X}\left[(\exp(-\hat{u}_\mathcal{S}(X^\mu)) - \exp(-u(X^\mu)))^2\right] \\
&\leq \mathbb{E}_{X^\mu \sim \mu_X}\left[(\hat{u}_\mathcal{S}(X^\mu) - u(X^\mu))^2\right] = \|\hat{u}_\mathcal{S} - u\|_{L^2(\mu_X)}^2.
\end{aligned}$$

Combining this with Lemma E.13 yields the desired result. $\qquad\square$

### E.3 CONVERGENCE RATES OF DENSITY RATIO ESTIMATOR

*Proof of Lemma 3.11.* According to Assumption 6 and Lemma 3.1, there exists $u \in \mathcal{U} = N(W_\mathcal{U}, L_\mathcal{U})$ such that

$$\inf_{u \in \mathcal{U}} \|u + \log \varrho\|_{L^2(Q_X)}^2 + \|u + \log \varrho\|_{L^2(P_X)}^2 \leq C_1(U_\mathcal{U} N_\mathcal{U})^{-4\alpha/d},$$

where $W = \mathcal{O}(U_\mathcal{U} \log U_\mathcal{U})$, $L = \mathcal{O}(N_\mathcal{U} \log N_\mathcal{U})$ and $C_1$ is a constant only depending on $\|\log \varrho\|_{\mathcal{H}^\alpha(\mathcal{X})}$, $d$ and $\alpha$. Using Lemma 3.2, we find that $\mathrm{VCdim}(\mathcal{U}) \leq C_2 U_\mathcal{U}^2 N_\mathcal{U}^2 (\log U_\mathcal{U} \log N_\mathcal{U})^2$, where the constant $C_2$ depends on $B$, $d$ and $\alpha$. By setting $U_\mathcal{U} N_\mathcal{U} = \mathcal{O}(n^{\frac{d}{2d+4\alpha}})$, we conclude the final result. $\qquad\square$

