# OpenReview forum: "On Convergence Rates of Deep Nonparametric Regression under Covariate Shift"
_ICLR.cc/2024/Conference — ICLR 2024 Conference Withdrawn Submission_

### Official Review · Reviewer_Wbn8 · 2023-10-29

**Soundness:** 3 good
**Presentation:** 3 good
**Contribution:** 2 fair
**Rating:** 5
**Confidence:** 3

**Summary:**

The paper proposed an adjusted nonparametric estimator that handles the problem of covariate shift. The new estimator is shown to outperform the estimator without adjustment. Specifically, this paper presents theoretical upper bound for the adjusted estimator, which differ from the bound of the un-adjusted estimator by a factor that is generally less than one, whose magnitude depends on the smoothness of the function to be estimated and the dimension of the covariate.

**Strengths:**

+ Careful mathematical analysis
+ The estimator looks novel

**Weaknesses:**

- No empirical analysis. Although this paper is highly mathematical, it could have showcased the proposed estimator’s performance with some empirical experiments. A paper without any empirical analysis is probably uncommon for a conference like ICLR

**Questions:**

1. The main estimator (2.10) is under the assumption that the data used to estimate the density ratio are independent from the data used to estimator the regression function $f$. Another estimator, perhaps more natural, is to estimate the density ratio and the regression function $f$ using “the same” $X$. That is, $\{X_i^\mu\}_i = \{X_i^P\}_i$. Could the current theory be modified to analyze the estimator under this setting?

2. [Minor] Some notations are not defined: Notations like dimensionality $d$, functional class $\mathcal H$ should have been defined where they first appear rather.

(1) What is the function class $\mathcal L$ for $u$ in Eq. (2.6).

(2) Lemma 2.1 and Remark 2.2 should be existing results? Please include reference for them.

3. The rate in Theorem 3.4 v.s. Theorem 3.8: from a bias variance decomposition perspective, is the convergence rate difference in the two theorems main due to the variance part? I would guess the mean of the estimators, reweighted or not, are the same.

---

### Official Review · Reviewer_GV9c · 2023-11-01

**Soundness:** 3 good
**Presentation:** 3 good
**Contribution:** 3 good
**Rating:** 6
**Confidence:** 3

**Summary:**

The paper aims to tackle the challenges arising from distribution mismatches in deep nonparametric regression. To address this, the authors introduce a two-stage pre-training reweighted framework that utilizes deep ReLU neural networks. They conduct a rigorous analysis of the convergence rates for three estimators: unweighted, reweighted, and pre-training reweighted. The analysis emphasizes the importance of the density ratio reweighting strategy in mitigating the impact of covariate shift.

**Strengths:**

1. By incorporating deep ReLU neural networks, the authors provide a novel methodology that can effectively handle covariate shift scenarios.

2. The study establishes fast convergence rates, indicating efficient learning and improved estimation performance.

3. The paper sheds light on the significance of the density-ratio reweighting strategy.

**Weaknesses:**

The paper is technically sound, but the limitations and drawback of the proposed methods are not clearly stated. A more comprehensive and detailed comparison between the proposed method and alternative approaches is necessary to provide a thorough evaluation.

**Questions:**

1. Assumption 2 seems to be a little bit strong. Is it possible to weaken it to make it more applicable? When considering $\Lambda$ as a finite constant, it can be observed that the convergence rates for the three different algorithms are equivalent.

2. Do the convergence rates achieve minimax optimality with respect to the parameter $\Lambda$?

---

### Official Review · Reviewer_FH47 · 2023-11-06

**Soundness:** 3 good
**Presentation:** 3 good
**Contribution:** 1 poor
**Rating:** 3
**Confidence:** 3

**Summary:**

This paper proposed some training schemes to deal with the covariate shift problem in deep learning, including re-weighted training and pre-training. Theoretically, convergence rates were derived.

**Strengths:**

Provided detailed mathematical derivations, and showed seemingly correct convergence rates. The analysis is based on ReLU neural networks.

**Weaknesses:**

1. The settings are restrictive, such as the accessibility to testing data (even without labels), and the bounded density ratio. Some papers reviewed did consider unknown testing distributions (no accessibility to testing data), such as Duchi & Namkoong (2021), Krueger et al. (2021), and Xu et al. (2022). Some others did consider unbounded density ratio, such as Ma et al. (2023).

2. There are no experimental studies. I know that some statistical papers have no experimental studies, but as a machine learning researcher, I think it is important to show that the methodology really works and to show the issues one may encounter in practice. The paper stated that it can offer guidance for practitioners.

**Questions:**

1. Because the paper assumed the accessibility to testing data without labels, the problem now becomes one type of domain adaptation. What is the advantage of such a methodology over popular domain adaptation methodologies?

2. Sun et al. (2011) was a much earlier work that adopted re-weighting samples. What is the difference from their method? Can the performance be compared?

3. I appreciate that the theoretical results with heavy mathematics are derived, but the paper can benefit from both simulation and empirical studies.